# *Arabidopsis* FORGETTER1 mediates stress-induced chromatin memory through nucleosome remodeling

Krzysztof Brzezinka[1†], Simone Altmann[1†‡], Hjördis Czesnick[1], Philippe Nicolas[1§], Michal Gorka[2], Eileen Benke[1], Tina Kabelitz[1], Felix Jähne[1], Alexander Graf[2], Christian Kappel[1], Isabel Bäurle[1*]

[1]Institute for Biochemistry and Biology, University of Potsdam, Potsdam, Germany; [2]Max-Planck-Institute for Molecular Plant Physiology, Potsdam, Germany

**Abstract** Plants as sessile organisms can adapt to environmental stress to mitigate its adverse effects. As part of such adaptation they maintain an active memory of heat stress for several days that promotes a more efficient response to recurring stress. We show that this heat stress memory requires the activity of the *FORGETTER1 (FGT1)* locus, with *fgt1* mutants displaying reduced maintenance of heat-induced gene expression. *FGT1* encodes the *Arabidopsis thaliana* orthologue of Strawberry notch (Sno), and the protein globally associates with the promoter regions of actively expressed genes in a heat-dependent fashion. FGT1 interacts with chromatin remodelers of the SWI/SNF and ISWI families, which also display reduced heat stress memory. Genomic targets of the BRM remodeler overlap significantly with FGT1 targets. Accordingly, nucleosome dynamics at loci with altered maintenance of heat-induced expression are affected in *fgt1*. Together, our results suggest that by modulating nucleosome occupancy, FGT1 mediates stress-induced chromatin memory.

*For correspondence: isabel. baeurle@uni-potsdam.de

[†]These authors contributed equally to this work

Present address: [‡]School of Life Sciences, University of Dundee, Dundee, United Kingdom; [§]Boyce Thompson Institute, Cornell University, Ithaca, United States

Competing interests: The authors declare that no competing interests exist.

## Introduction

Abiotic stress is a major threat to global crop yields and this problem is likely to be exacerbated in the future. A large body of research has focused on the immediate stress responses. However, in nature, stress is frequently chronic or recurring, suggesting that temporal dynamics are an important, but under-researched, component of plant stress responses. Indeed, plants can be primed by a stress exposure such that they respond more efficiently to another stress incident that occurs after a stressless period (*Hilker et al., 2015*). Priming has been described in response to pathogen attack, heat stress (HS), drought, and salt stress (*Charng et al., 2006*; *Conrath, 2011*; *Jaskiewicz et al., 2011*; *Ding et al., 2012*; *Sani et al., 2013*). Such stress priming and memory may be particularly beneficial to plants due to their sessile life style.

Chromatin structure can be modulated by nucleosome positioning, histone variants and post-translational histone modifications that together control the access of sequence-specific transcription factors and the general transcription machinery to gene loci (*Struhl and Segal, 2013*; *Zentner and Henikoff, 2013*). Chromatin mediates long-term stability of environmentally and developmentally-induced gene expression states (*Gendrel and Heard, 2014*; *Steffen and Ringrose, 2014*; *Berry and Dean, 2015*). Hence, the modification of chromatin structure has been suggested to mediate the priming and memory of stress-induced gene expression. Indeed, the above mentioned cases of plant stress priming are associated with lasting histone H3 methylation (*Conrath, 2011*; *Jaskiewicz et al., 2011*; *Ding et al., 2012*; *Sani et al., 2013*; *Lämke et al., 2016*). However, the

**eLife digest** In nature, plant growth is often limited by unfavourable conditions or disease. Plants have thus evolved sophisticated mechanisms to adapt to such stresses. In fact, brief exposure to stress can prime plants to be better prepared for a future stress following a period without stress. However, the molecular basis of this memory-like phenomenon is poorly understood.

Now, Brzezinka, Altmann et al. have used priming by heat stress as a model to dissect the memory of environmental stresses in thale cress, *Arabidopsis thaliana*. First, a library of mutant plants were tested to identify a gene that is specifically required for heat stress memory but not for the initial responses to heat. Brzezinka, Altmann et al. identified one such gene and termed it *FORGETTER1* (or *FGT1* for short). Further experiments then revealed that the FGT1 protein binds directly to a specific class of heat-inducible genes that are relevant for heat stress memory.

Brzezinka, Altmann et al. propose that the FGT1 protein makes sure that the heat-inducible genes are always accessible and active by modifying the way the DNA containing these genes is packaged. DNA is wrapped around protein complexes called nucleosomes and depending on how tightly the DNA of a gene is wrapped makes it more or less easy to activate the gene. In agreement with this model, FGT1 does interact with proteins that can reposition nucleosomes and leave the DNA more loosely packaged. Also, the fact that plants that lack a working *FGT1* gene repackage the DNA of memory-related genes too early after experiencing heat stress provides further support for the model.

Together these findings could lead to new approaches for breeding programmes to improve stress tolerance in crop plants. One future challenge will be to find out whether memories involving nucleosomes are also made in response to other stressful conditions, such as attack by pests and disease.

underlying mechanism and the contribution of other determinants of chromatin structure such as nucleosome positioning and occupancy remain unknown.

Moderate HS allows a plant to acquire thermotolerance and subsequently withstand high temperatures that are lethal to a plant in the naïve state (*Mittler et al., 2012*). After returning to non-stress temperatures, acquired thermotolerance is maintained over several days, and this maintenance phase is genetically separable from the acquisition phase (*Charng et al., 2006*, *2007*; *Meiri and Breiman, 2009*; *Stief et al., 2014*). We refer to this maintenance phase as HS memory. The immediate HS response (acquisition phase) involves the activation of heat shock transcription factors (HSFs) that induce heat shock proteins (HSPs). Their chaperone activity ensures protein homeostasis (*Scharf et al., 2012*). The HS response is conserved in animals, plants and fungi (*Richter et al., 2010*).

Among the 21 HSFs in *Arabidopsis thaliana* (*Scharf et al., 2012*), only *HSFA2* is specifically required for HS memory (*Charng et al., 2007*). It activates *HEAT-STRESS-ASSOCIATED32* (*HSA32*), a gene with chaperone-like activity, although no homology to known chaperone families (*Wu et al., 2013*). Like *HSFA2*, *HSA32* is critically required for HS memory (*Charng et al., 2006*). *HSA32* is induced by HS and this induction is sustained for at least three days. A set of HS memory-related genes was identified based on their similar expression pattern, which is in contrast to that of canonical HS-inducible genes (*HSP70*, *HSP101*) that show upregulation after HS, but not sustained induction (*Stief et al., 2014*). Among the HS memory-related genes are small HSPs (such as *HSP21*, *HSP22.0*, *HSP18.2*). A subset of these loci show transcriptional memory in the sense that recurring stress causes a more efficient re-activation compared to the first stress incident, even though active transcription has subsided before the second stress (*Lämke et al., 2016*). Sustained induction and transcriptional memory of these genes is associated with hyper-methylation of H3K4 (H3K4me2 and H3K4me3) and requires HSFA2, which binds directly to these genes (*Lämke et al., 2016*). Interestingly, HSFA2 dissociates from these loci before its requirement becomes apparent at the physiological and gene expression levels, thus implicating the existence of additional factors.

Here, we report the identification of *FGT1* from an unbiased screen for factors that are required for the sustained induction of *HSA32*. *FGT1* is required for HS memory at the physiological and

gene expression levels. FGT1 is the single *A. thaliana* orthologue of metazoan Strawberry notch, a highly conserved co-activator of the developmental regulator Notch. We show that FGT1 associates with memory genes in a HS-dependent way. Moreover, FGT1 is widely associated with the transcriptional start site of expressed genes. We further show that FGT1 interacts with highly conserved chromatin remodeling complexes and is required for proper nucleosome dynamics at HS-memory genes. Thus, FGT1 maintains its target loci in an open and transcription-competent state by interacting with remodeler complexes around the transcriptional start site.

## Results

### *FGT1* is required for HS memory and sustained induction of *HSA32* and other memory genes

In order to identify regulators of HS memory we generated a transgenic *HSA32::HSA32-LUCIFERASE* (*HSA32::HSA32-LUC*) reporter line. *LUC* expression in this line was induced by HS and expression remained high for at least 3 d (*Figure 1A*), thus mimicking expression of the endogenous *HSA32* (*Charng et al., 2006*). We mutagenized the *HSA32::HSA32-LUC* line with ethyl methanesulfonate and screened M2 families for mutants with modified maintenance of LUC activity after HS. To this end, 4 d-old plate-grown seedlings were treated with an acclimatizing heat treatment (ACC, see Materials and methods). LUC-derived bioluminescence was monitored 1, 2 and 3 d later, and putative mutants were isolated that had normal LUC activity 1 d after ACC and reduced activity 3 d after ACC. Among the recovered mutants with such a *LUC* expression profile was *forgetter1* (*fgt1-1*), on which we focused further analyses. *LUC* expression in *fgt1-1* was induced normally, however, it declined precociously, which was most apparent 3 d after ACC.

We next investigated whether this correlated with modified HS memory at the physiological level by applying a tester HS 2 or 3 d after ACC. The tester HS is lethal to a naïve plant or a mutant with loss of HS memory. Indeed, *fgt1-1* mutants displayed reduced growth and survival under these conditions (*Figure 1B–D*). To check whether *fgt1-1* mutants had a generally impaired HS response, we also tested *fgt1-1* seedlings for acquisition of thermotolerance and for basal thermotolerance, i. e. the amount of heat that can be tolerated without prior acclimation. *fgt1-1* mutants behaved very similar to the parental line in these assays (*Figure 1—figure supplements 1,2*), indicating that the immediate responses to HS were not affected in *fgt1-1*. Thus, *fgt1-1* is specifically impaired in HS memory. Notably, *fgt1-1* did not have any obvious morphological alterations under standard growth conditions.

We next examined whether the premature decline of *LUC* expression mimicked that of the endogenous *HSA32* gene by quantitative RT-PCR (qRT-PCR) specific for the endogenous *HSA32*. *HSA32endo* and *LUC* transcripts were induced normally in *fgt1-1* during the first day after ACC, but declined faster thereafter (*Figure 1E*, *Figure 1—figure supplement 3*). A similar defect was observed for the HS memory-related genes *HSP21*, *HSP22.0* and *HSP18.2*, but not the HS-inducible non-memory genes *HSP101* and *HSP70*. For intron-containing genes, we measured unspliced transcripts as a proxy for transcriptional activity. Unspliced *HSA32* transcripts were induced in *fgt1-1* similarly as in the parent up to 21 h, but declined faster thereafter. Notably, unspliced transcript levels in the parent were still elevated 20-fold relative to the non-HS control (NHS) at 69 h after ACC, indicating continued transcription throughout the memory phase. Similar results were obtained for *HSP21*, but not for the non-memory genes. This is in accordance with what was observed previously (*Lämke et al., 2016*). Thus, *FGT1* is required to facilitate sustained transcription of HS memory genes after HS.

### *FGT1* encodes the orthologue of *Drosophila* Strawberry notch, a protein involved in induction of Notch and EGFR target genes

To identify the molecular lesion underlying the *fgt1-1* mutant phenotype, we combined recombination breakpoint mapping with Illumina sequencing. We identified a 0.77 MB interval at the bottom of chromosome 1, containing a splice acceptor site mutation in *At1g79350* (*Figure 2A*). This mutation caused retention of intron 19, resulting in a premature stop codon. *At1g79350* was previously tentatively identified as *EMB1135* with a reported embryo-defective phenotype (*Meinke et al., 2008*). However, it was not confirmed that the disruption of this gene in the *emb1135* allele indeed

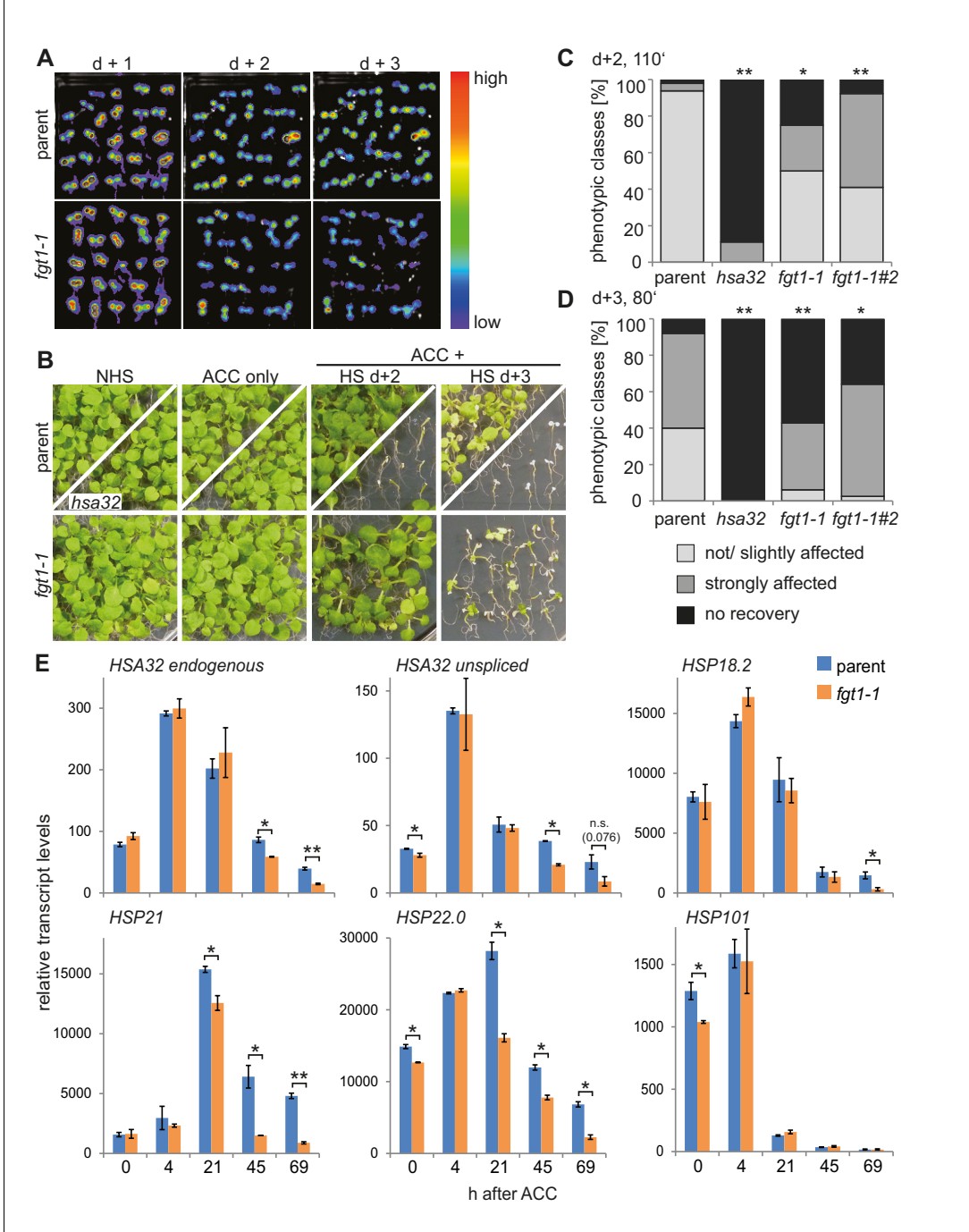

**Figure 1.** *FGT1* is required for HS memory and sustained induction of memory genes in *A. thaliana*. (A) *fgt1-1* displays normal induction but reduced maintenance of *pHSA32::HSA32-LUC* expression. Bioluminescence of *fgt1-1* or the parent assayed 1, 2, or 3 d after an acclimatizing HS (ACC). The color scale of relative LUC activity is shown. (B) *fgt1-1* is impaired in HS memory at the physiological level. Seedlings of the indicated genotypes (cf. *fgt1-1#2* in C–D) were acclimatized 4 d after germination and treated with a tester HS 2 or 3 d later. Pictures were taken after 14 d of recovery. (C–D) Quantification of the data shown in (B). The *fgt1-1* lines represent independent backcrosses. Data are averaged over at least two independent assays (n>36). Fisher's exact test, *p<0.05; **p<0.001. (E) Transcript levels of HS memory genes after ACC decline prematurely in *fgt1-1*. Expression values were normalized to the reference *At4g26410* and the corresponding no-HS control (NHS). Data are averages and SE of two biological replicates. *p<0.05; **p<0.01 (Student's t test).

The following figure supplements are available for figure 1:

**Figure supplement 1.** *fgt1-1* is not affected in the acquisition of thermotolerance.

*Figure 1 continued on next page*

*Figure 1 continued*

**Figure supplement 2.** *fgt1-1* has normal basal thermotolerance.

**Figure supplement 3.** Additional qRT-PCR analyses of HS genes.

causes the phenotype and neither *fgt1-1*, nor any of several putative loss-of-function T-DNA insertion lines showed any obvious morphological phenotype. Three independent lines of evidence show that *FGT1* is *At1g79350*. First, we complemented the *LUC* expression and physiological memory phenotypes by expressing a genomic *FGT1* fragment (SA13) in the *fgt1-1* background (*Figure 2B– D*). Second, similar results were obtained for a FGT1-YFP fusion protein that was driven by the constitutive *35S* CaMV promoter (*Figure 2—figure supplement 1A,B*). Finally, *fgt1-2* and *fgt1-3*, two putative loss-of-function T-DNA alleles displayed reduced HS memory (*Figure 2—figure supplement 1C–E*).

FGT1 contains an ATP-binding DExD/H-like helicase domain, a Helicase C-like domain, and a PHD finger (*Figure 2A*). *FGT1* is a single copy gene in *A. thaliana*. Interestingly, it is highly homologous over the whole length of the protein with Sno from *Drosophila melanogaster*, human SBNO1 and SBNO2, and *Caenorhabditis elegans let-765* (*Figure 2—figure supplement 2*) (*Majumdar et al., 1997*; *Simms and Baillie, 2010*; *Grill et al., 2015*). *Sno* genes are required for the expression of Notch and EGFR target genes and it has been hypothesized that they interact with co-activator proteins to spatio-temporally regulate transcription (*Tsuda et al., 2002*), yet no molecular mode of action has been demonstrated. Although DExD helicases have been ascribed a role in RNA processing and translation, roles in gene expression and transcription have been suggested (*Fuller-Pace, 2006*). *FGT1* is expressed throughout the plant (*Winter et al., 2007*) and was slightly induced (1.7 fold) at 4 h after HS, but not thereafter (*Figure 2E*).

We next tested the subcellular localization of the complementing FGT1-YFP fusion protein in roots of 3 d-old stably transformed seedlings. FGT1-YFP was localized to the nucleus (*Figure 2F*). PHD domains have the potential to bind to methylated histone H3 tails (*Musselman and Kutateladze, 2011*). We thus tested whether FGT1$_{PHD}$-GST was precipitated by H3 histone tail peptides that were either unmethylated or mono-, di-, or trimethylated, respectively (*Figure 2—figure supplement 3*). We observed comparable binding to H3 aa 1–20, H3 aa 21–44 or H3 aa 1–20 methylated at K4 or K9. In contrast, the PHD domain of ING1 (*Lee et al., 2009*) bound under the same conditions only to the H3K4me3 peptide (*Figure 2—figure supplement 3*). This suggests that FGT1-PHD binds to the N-terminal region of H3, albeit not in a methylation-specific manner (at least with respect to methylated K4 and K9). In addition, histone H3 was co-immunoprecipitated with FGT1-YFP, but not YFP alone, from extracts of transgenic *A. thaliana* seedlings (*Figure 2G*). In summary, nuclear FGT1 is the *A. thaliana* orthologue of the DExD helicase Sno and associates with H3 in vivo, consistent with a function as a co-activator.

## FGT1 associates with *HSA32* and other memory genes

Given its potential function as a co-activator, we next asked whether FGT1 binds directly to its putative target genes during HS memory. To test this, we performed chromatin immunoprecipitation (ChIP) followed by qPCR analysis on *35S::FGT1-YFP* plants (*Figure 3*). FGT1 bound to a broad region around the transcriptional start sites (TSS) of *HSA32, HSP18.2*, and HSP22.*0*. The enrichment of FGT1 in the heat-treated samples was highest 4 h and 28 h after ACC compared to the NHS, and was still present at 52 h. Such heat-dependent enrichment was not observed at the *ACTIN7 (ACT7)* and *AtMu1* control loci. The enrichment at the active *ACT7* gene was comparable to that of *HSA32* before HS, suggesting that FGT1 binds *HSA32* and *ACT7* already pre-HS. The ChIP-signal in FGT1-YFP plants at bound loci was strongly enhanced compared to non-transgenic control samples (*Figure 3—figure supplement 1A*). A comparable but overall weaker binding pattern was observed for a FGT1-YFP driven by the endogenous promoter (*Figure 3—figure supplement 1B,C*), indicating that the observed binding pattern does not result from *FGT1* overexpression. Thus, FGT1 binds to

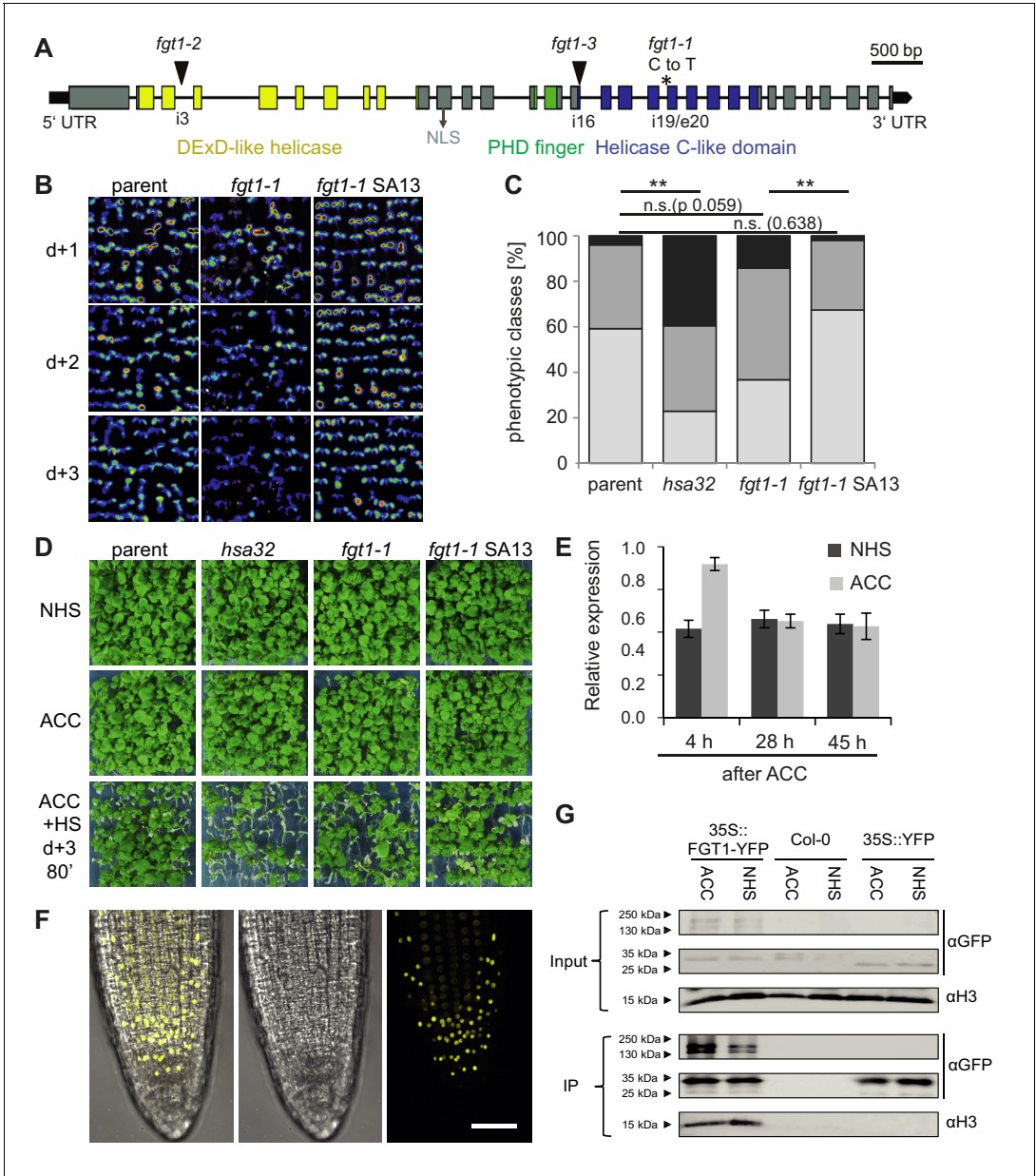

**Figure 2.** *FGT1* encodes the *A. thaliana* orthologue of *Drosophila* Sno and binds histones. (**A**) Gene model of *FGT1* (*At1g79350*) with domains and location of mutations; exons (grey and colored bars); black line, intron. *fgt1-1* has a C to T mutation at the splice acceptor site of intron 19/exon 20. (**B**–**D**) Complementation of *fgt1-1* by a genomic *FGT1* fragment (SA13). (**B**) pHSA32::HSA32-LUC-derived bioluminescence of indicated genotypes assayed 1, 2, or 3 d after ACC. (**C,D**) Seedlings of the indicated genotypes were acclimatized 5 d after germination and received a tester HS 3 d later. (**C**) Quantification; n = 48–49, Fisher's exact test, **p<0.01. (**D**) Representative picture taken after 14 d recovery. (**E**) *FGT1* transcript levels increase transiently after ACC. Relative *FGT1* transcript levels were determined by qRT-PCR and normalized to *At4g26410*. Errors are SE of two biological replicates. (**F**) FGT1 is localized to the nucleus in 3 d-old seedling roots. *35S::FGT1-YFP* transgenic seedlings were imaged for YFP fluorescence. Left, overlay; middle, bright field; right, YFP fluorescence. Scale bar, 40 µm. (**G**) FGT1 binds histone H3 *in vivo*. Nuclear protein extracts of transgenic *35S::FGT1-YFP, 35S::YFP* and non-transgenic Col-0 seedlings harvested 28 h after the indicated treatments were immuno-precipitated with anti-GFP antibody. Co-purification of histone H3 was assessed by immunoblotting.

The following figure supplements are available for figure 2:

**Figure supplement 1.** Complementation of the *fgt1-1* mutant phenotype during HS memory.

**Figure supplement 2.** FGT1 is highly conserved and encodes the SNO/SBNO orthologue of *A. thaliana*.

*Figure 2 continued on next page*

*Figure 2 continued*

**Figure supplement 3.** The PHD domain of FGT1 binds to histone H3 *in vitro*.

memory genes in a region encompassing the TSS and proximal promoter, where it may mediate their sustained expression after HS.

## FGT1 binds widely to the proximal promoter of expressed genes

Because of the highly conserved nature of FGT1 and the binding to *ACT7*, we suspected that FGT1 may have genomic targets beyond the tested candidates. To obtain a global view, we performed ChIP-seq on *35S::FGT1-YFP* plants 28 h after ACC or NHS. Peak calling identified 942 (60) genes

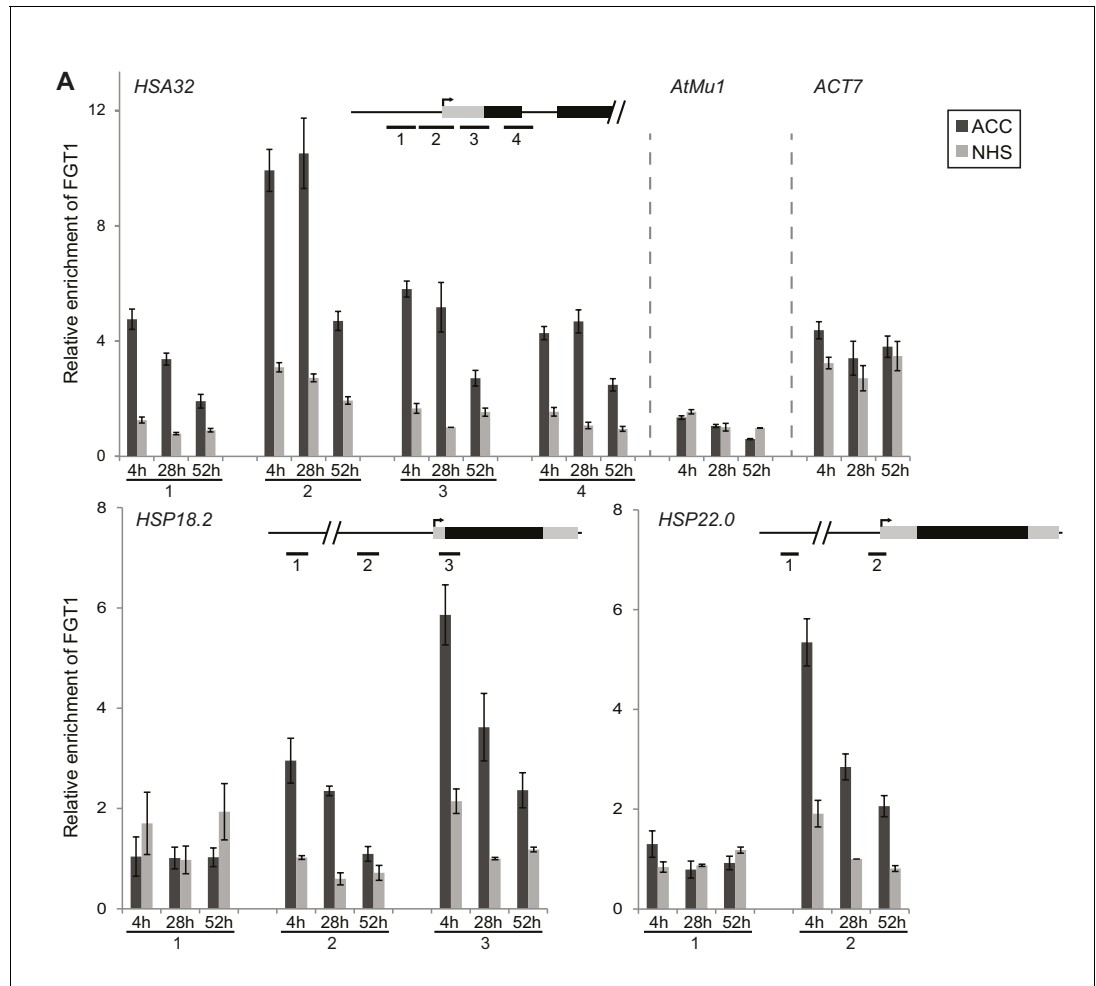

**Figure 3.** FGT1 binds memory-associated genes in a HS-dependent manner. FGT1 binds to *HSA32*, *HSP18.2* and *HSP22.0*. ChIP-qPCR on *35S::FGT1-YFP* seedlings was performed 4, 28 or 52 h after an acclimatizing HS (ACC) or no HS (NHS). As controls, the active gene *ACT7* or the inactive transposon *AtMu1* were used. Schematics show regions analyzed relative to TSS. *HSA32* 1–4: −175, −75, +57, +194 bp. *HSP18.2* 1–3: −1068, −367, +32 bp. *HSP22.0* 1–2: −3000, −60 bp; *ACT7*: +55 bp. *AtMu1*: -175 bp relative to ATG. Amplification values were normalized to input and region $2_{HSA32}$ at 28 h NHS (*HSA32*, *AtMu1* and *ACT7*), region $2_{HSP18.2}$ at 28 h NHS or region $3_{HSP22.0}$ at 28 h NHS, respectively. Data are averages of at least three biological replicates. Error bars indicate SE.

The following figure supplement is available for figure 3:

**Figure supplement 1.** FGT1-YFP expressed from the endogenous promoter binds to *HSA32*.

with FGT1 enrichment after ACC (NHS), and no binding in the corresponding non-transgenic control samples. While the NHS peaks remained associated with FGT1 after ACC, the ACC peaks were overall less strongly enriched under NHS conditions (*Figure 4—figure supplement 1*). Coverage profiling of the FGT1-associated genes indicated that FGT1 bound primarily to the proximal promoter just upstream of the TSS and somewhat more weakly to the region downstream of the transcription termination site (TTS). In contrast, the signal was very low in the transcribed region. For both conditions (ACC and NHS), FGT1-associated genes showed a higher expression in seedlings under normal growth conditions compared to non-target genes (*Figure 4A*), suggesting that FGT1 binding is positively correlated with transcription.

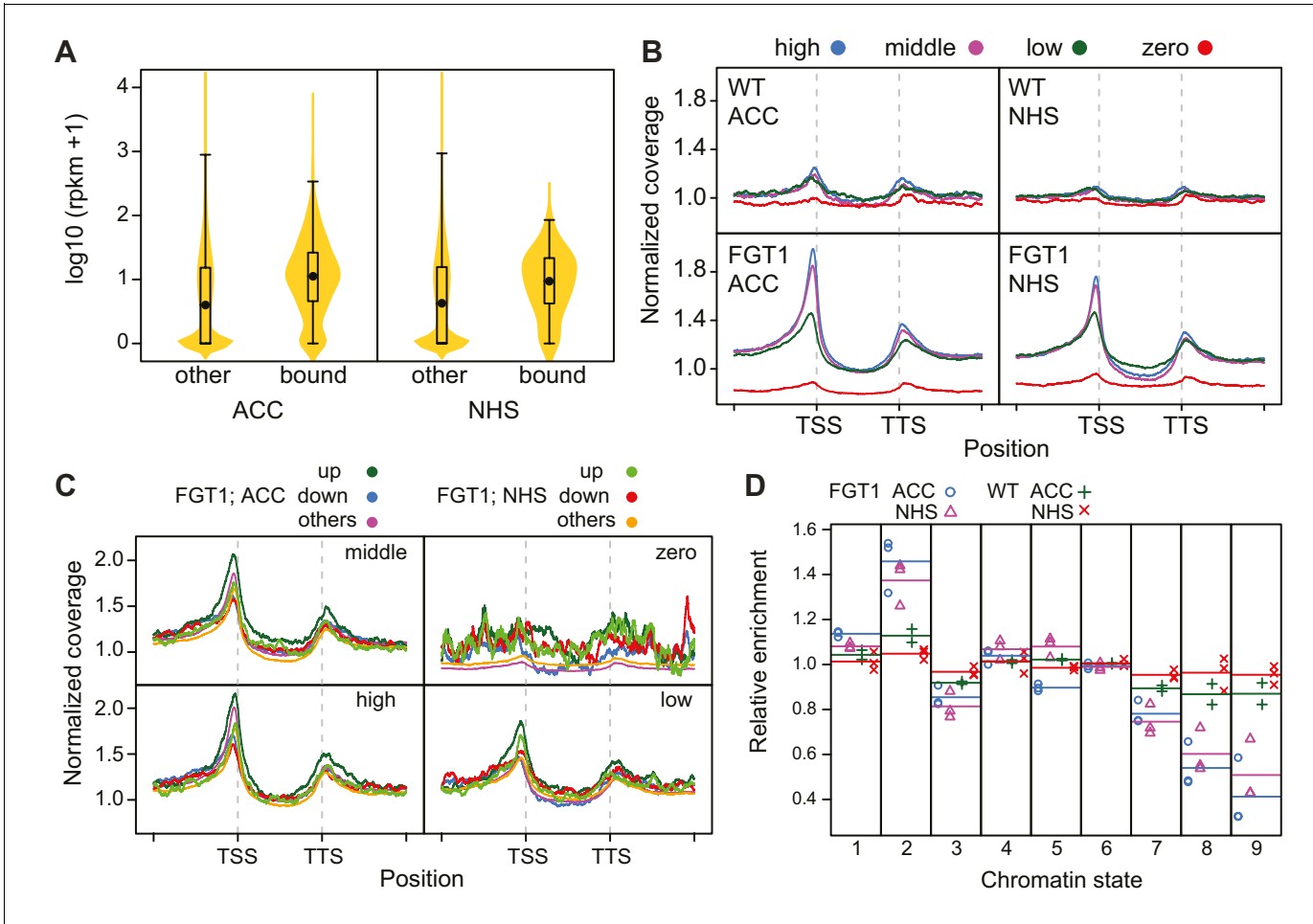

**Figure 4.** FGT1 globally binds expressed genes upstream of the TSS. (**A**) FGT1-associated genes (ChIP-seq peaks) are more highly expressed than other genes. Violin plot indicates expression levels of FGT1-bound genes compared to all other genes (*Figure 4—figure supplement 1*). Expression data were taken from (*Gan et al., 2011*). (**B**) Normalized global read coverages of ACC or NHS FGT1-YFP or wild-type control samples as determined by ChIP-seq. Genes were categorized into not expressed genes and equally sized groups of highly, moderately and lowly expressed genes (*Gan et al., 2011*). Coverage profiles include 2 kb up- and downstream of the TSS and TTS, respectively. Genic regions were normalized to a standard length. (**C**) Normalized read coverages of HS-responsive genes from ACC or NHS FGT1-YFP. The panels show global analysis of genes in the respective expression class (cf. B) according to their expression pattern 4 h after ACC (*Stief et al., 2014*) in wild type (up, down, others). (**D**) FGT1 is enriched in chromatin state 2 (*Sequeira-Mendes et al., 2014*). Relative enrichment of FGT1-bound sequences or WT control after ACC or NHS in different chromatin states indicated depletion of FGT1 in heterochromatin (states 8, 9) and enrichment in state 2 (promoter and intergenic regions, open chromatin). Lines denote average of replicates.

The following figure supplement is available for figure 4:

**Figure supplement 1.** Coverage profiles of ChIP-seq peaks bound by FGT1 28 h after ACC or NHS.

Given that the identified peaks were wide and flat, we hypothesized that the peak calling may have underestimated the number of targets. Thus, we investigated the global correlation of FGT1 binding and expression under NHS conditions by plotting global coverage profiles grouped according to the relative expression in non-stressed seedlings (*Gan et al., 2011*). This revealed that FGT1 is preferentially associated with expressed genes at a global scale (*Figure 4B*). FGT1 is most strongly associated with genes that have high or intermediate expression in a pattern similar to that observed for the peak genes (*Figure 4—figure supplement 1*). We next asked whether FGT1 shows differential association with genes that are HS-responsive (*Stief et al., 2014*). FGT1 associated most strongly with those expressed genes that are upregulated at 4 h after ACC, irrespective of their expression level before HS (*Figure 4C*). This is especially true for genes that are lowly expressed without HS; this category contains typical HS-responsive genes. Accordingly, FGT1 associated most strongly with genes that are upregulated at 4 h and/ or 52 h after ACC and this is more pronounced in the ACC samples (Figure 7C). Thus, HS increases binding of FGT1 to HS-responsive genes globally, and these genes are associated with low levels of FGT1 already before HS.

The *A. thaliana* genome was categorized into nine chromatin states based on the differential presence of histone modifications, variants and DNA methylation (*Sequeira-Mendes et al., 2014*). Analyzing the overlap between FGT1-bound sequences with different chromatin states, we found that FGT1 was highly enriched in sequences annotated as chromatin state 2 (*Figure 4D*). This state is found in poised chromatin, mostly in promoters and intergenic regions (hence transcript levels are low). It is enriched in H3.3, H3K4me2, H3K4me3, H2A.Z, H2Bub, H3K27me3, AT-rich and relatively low in overall nucleosome abundance. Strikingly, state 2 peaks immediately before the TSS, and has a smaller peak just after the TTS, mimicking closely the global coverage profile of FGT1 (*Sequeira-Mendes et al., 2014*). In contrast, FGT1 was depleted from the heterochromatic states 8 and 9. Thus, FGT1 binding globally associates with the nucleosome-poor regions flanking the transcription units of expressed genes.

## FGT1 interacts with SWI/SNF and ISWI chromatin remodelers

To elucidate the mechanism of how FGT1 promotes gene expression, we isolated FGT1-interacting proteins. To this end, we purified native FGT1-YFP complexes from *35S::FGT1-YFP* seedlings 28 h after ACC or control (NHS) treatment. FGT1-YFP and associated proteins were then subjected to LC-MS/MS analysis. As controls, we performed purifications on *35S::YFP* and Col-0 plants, respectively. Among the peptides identified specifically in the FGT1-YFP samples were both *A. thaliana* orthologues of the ISWI chromatin remodeler, CHR11 and CHR17, and the SWI/SNF chromatin remodeler BRAHMA (BRM), suggesting that FGT1 interacts with chromatin remodeling proteins (*Table 1*). Because of the high homology between CHR11 and CHR17, most of the identified peptides could not be assigned unequivocally to either of the two proteins, however, a few specific peptides were recovered demonstrating the presence of both ISWI proteins (*Table 1*). We also identified several known subunits (SWI3a, b, d, SWP73b) of the BRM complex. We did not observe differences between the ACC and NHS samples, suggesting that the mode of action of FGT1 is independent of HS. To confirm the interactions between FGT1 and the remodelers we used bimolecular fluorescent complementation in transiently transformed tobacco leaves (*Walter et al., 2004*). We thus confirmed the interaction of FGT1 with CHR11, CHR17 and BRM in the nucleus (*Figure 5*). In summary, FGT1 interacts with chromatin remodeling proteins of the ISWI and SWI/SNF classes.

## BRM and ISWI are required for HS memory

To determine whether the interaction of FGT1 and the remodelers was functionally relevant during HS memory, we examined whether remodeler mutants displayed normal HS memory. As the loss of *BRM* causes sterility, we performed the assay on the progeny of a heterozygous *brm-1/+* plant and genotyped individual seedlings after phenotyping. Indeed, *brm-1* mutants displayed reduced HS memory (*Figure 6A,B*). The highly similar CHR11 and CHR17 proteins show functional redundancy and the double mutant displays severe developmental defects including dwarfism (*Li et al., 2012*). Thus, we performed the experiment on the progeny of a *chr11/chr11 chr17/+* plant. As for *brm-1*, we genotyped individual seedlings after the phenotypic analysis was completed. We observed that *chr11* single mutants and *chr11/chr11 chr17/+* seedlings were defective in the physiological HS memory (*Figure 6A,B*). Due to their growth defects, we could not analyze *chr11 chr17* double

**Table 1.** FGT1 interacts with chromatin remodeling proteins *in vivo*. FGT1-interacting proteins identified by native co-immunoprecipitation followed by mass spectrometry (nHPLC-MS/MS) from 5 d-old *35S::FGT1-YFP* seedlings subjected to ACC or NHS 28 h before sampling. Col-0 and *35S::YFP* were used as controls. The data represent the number of unique peptides found in the indicated experiments.

| Background | Treatment | Exp | Number of peptides | | | | | | | | |
|---|---|---|---|---|---|---|---|---|---|---|---|
| | | | FGT1 | CHR11/ CHR17 | Chr11 | Chr17 | BRM | SWI3a | SWI3b | SWI3d | SWP73b |
| 35S::FGT1-YFP | ACC | 1 | 58 | 4 | 1 | - | - | - | - | - | - |
| | | 2 | 56 | 12 | 3 | 2 | 2 | 3 | 1 | 1 | 3 |
| | | 3 | 43 | 11 | - | - | 2 | - | - | - | - |
| | NHS | 1 | 33 | 2 | - | - | - | - | - | - | 1 |
| | | 2 | 52 | 11 | 4 | 2 | 1 | 4 | - | - | 3 |
| | | 3 | 51 | 4 | - | - | - | - | - | - | - |
| Col-0 | ACC | 1-3 | - | - | - | - | - | - | - | - | - |
| | NHS | 1-3 | - | - | - | - | - | - | - | - | - |
| 35S::YFP | NHS | 1-3 | - | - | - | - | - | - | - | - | - |

mutants. To test whether the remodeler mutants have a generally impaired HS response, we also tested their ability to acquire thermotolerance and their basal thermotolerance. Mutants in the remodelers behaved similar to the wild type or slightly better in these assays (*Figure 6—figure supplements 1*, *2*), which indicates that the responses to acute HS were not compromised. Thus, the remodeler mutants under investigation displayed a specific impairment of HS memory and not a general defect in HS responses. We also tested the expression of HS-responsive genes after ACC in wild type, and *brm-1/+* or *chr11/chr11 chr17/+* -segregating lines, respectively. qRT-PCR analysis revealed that both mutant lines show a premature decline of expression of *HSA32, HSP18.2, HSP21, HSP22* and *HSP101* (*Figure 6C*). In many cases, transcript levels were already lower at the earliest time point measured (immediately after the end of ACC), suggesting that the remodelers were also necessary for full induction of these genes. Interestingly, this was not correlated with a reduced level of acquired thermotolerance in our assays (*Figure 6—figure supplement 1*).

### Genetic interaction of *FGT1* and *BRM*

Given the similar HS memory phenotypes and their physical interaction, we asked whether *FGT1* interacted with *BRM* also genetically. Indeed, the *brm-1 fgt1-1* double mutants displayed several novel phenotypes compared to the *brm-1* single mutant (*Figure 6D,E*). During seedling development, growth of the *brm-1 fgt1-1* double mutant was retarded, resulting in reduced development and delayed leaf initiation (*Figure 6—figure supplement 3*). Later on, flowering time (days to flowering) was delayed compared to the wild type or either single mutant (*Figure 6D,E*). As the double mutants have to be isolated from a segregating population due to the sterility of *brm-1*, we could not obtain suitable material for testing HS memory. Nevertheless, the additive phenotypes observed in the double mutant clearly suggest that both genes act partially redundantly, in agreement with the idea that they act at least partly on the same targets. The additional phenotypes indicate that *FGT1* also plays a role during plant development, which is consistent with the large number of target genes that are not related to HS memory.

### BRM globally binds HS memory genes and FGT1 target genes

To compare BRM and FGT1 genomic targets, we took advantage of published BRM ChIP-seq data (*Li et al., 2016*). We observed a highly significant overlap between genes associated with BRM under non-stress conditions and FGT1-associated genes (NHS and ACC; *Figure 7A*). Globally, BRM was more strongly associated with FGT1 ACC target genes than with the rest of the genome (*Figure 7B*). Strikingly, BRM showed a very similar coverage profile as FGT1; a pronounced peak at the TSS and a second, weaker peak just downstream the TTS. When comparing the association with genes that are induced at 4 and/or 52 h after ACC, we found BRM to be strongly enriched at

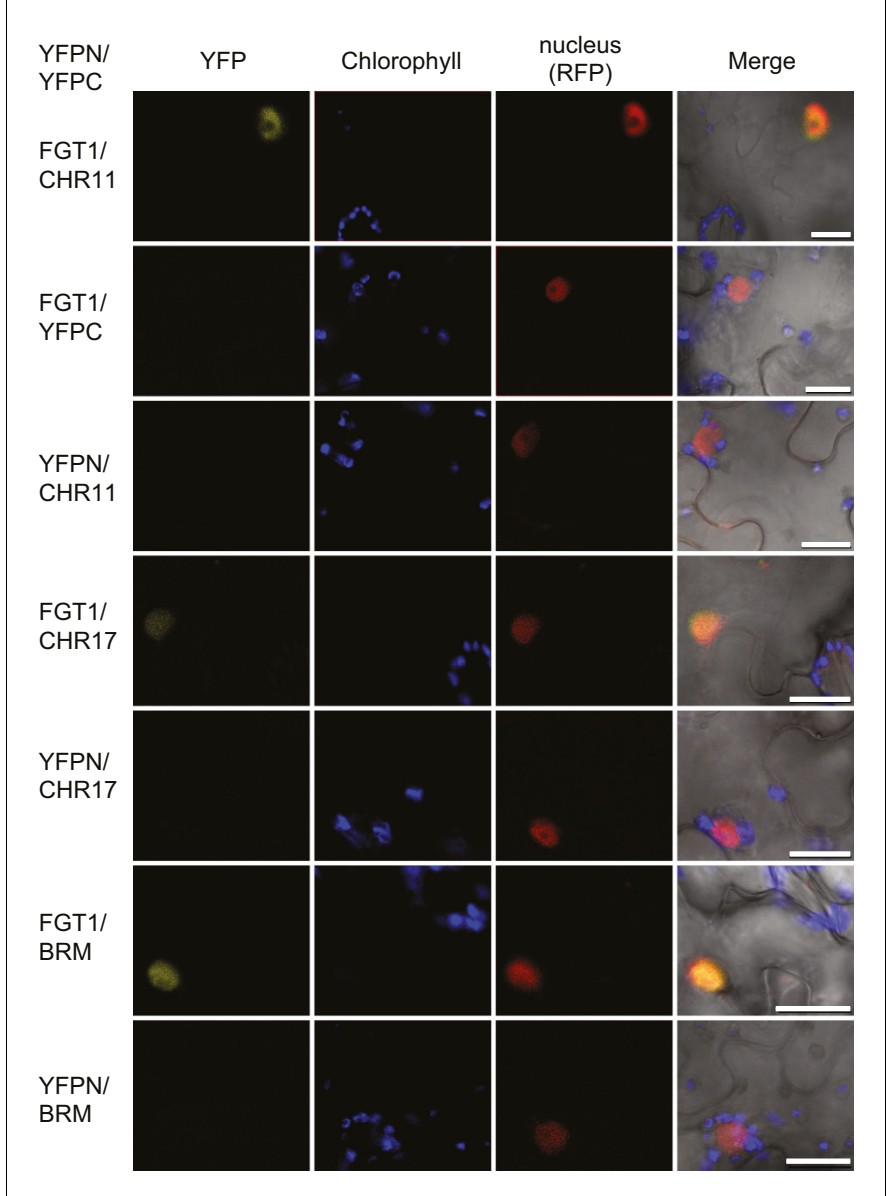

**Figure 5.** FGT1 interacts *in vivo* with SWI/SNF (BRM) and ISWI (CHR11, CHR17) chromatin remodeling proteins. Bimolecular Fluorescence Complementation confirms the interaction of FGT1 and CHR11, CHR17 or BRM in the nucleus of tobacco leaf cells. The indicated constructs were co-transformed and analyzed 2 d later with an LSM710 confocal microscope. YFP, BiFC signal in the YFP spectrum; RFP, signal from co-expressed nuclear RFP-fusion protein. Size bar, 20 μm.

memory genes (up at 4 and 52 h) and late-inducible genes (up at 52 h only), compared to downregulated or non-responsive genes (*Figure 7C*). Again, this was very similar to the results observed for FGT1. Notably, the association of BRM and (to a lesser extent) FGT1 with memory genes was established before HS. The overlapping binding pattern of FGT1 and BRM was also apparent from browser screenshots of individual loci (*Figure 7—figure supplement 1*). In summary, genome-wide BRM target genes overlap strongly with FGT1 target genes and they are pre-associated with memory genes under non-stress conditions, strongly supporting the interaction of FGT1 and BRM.

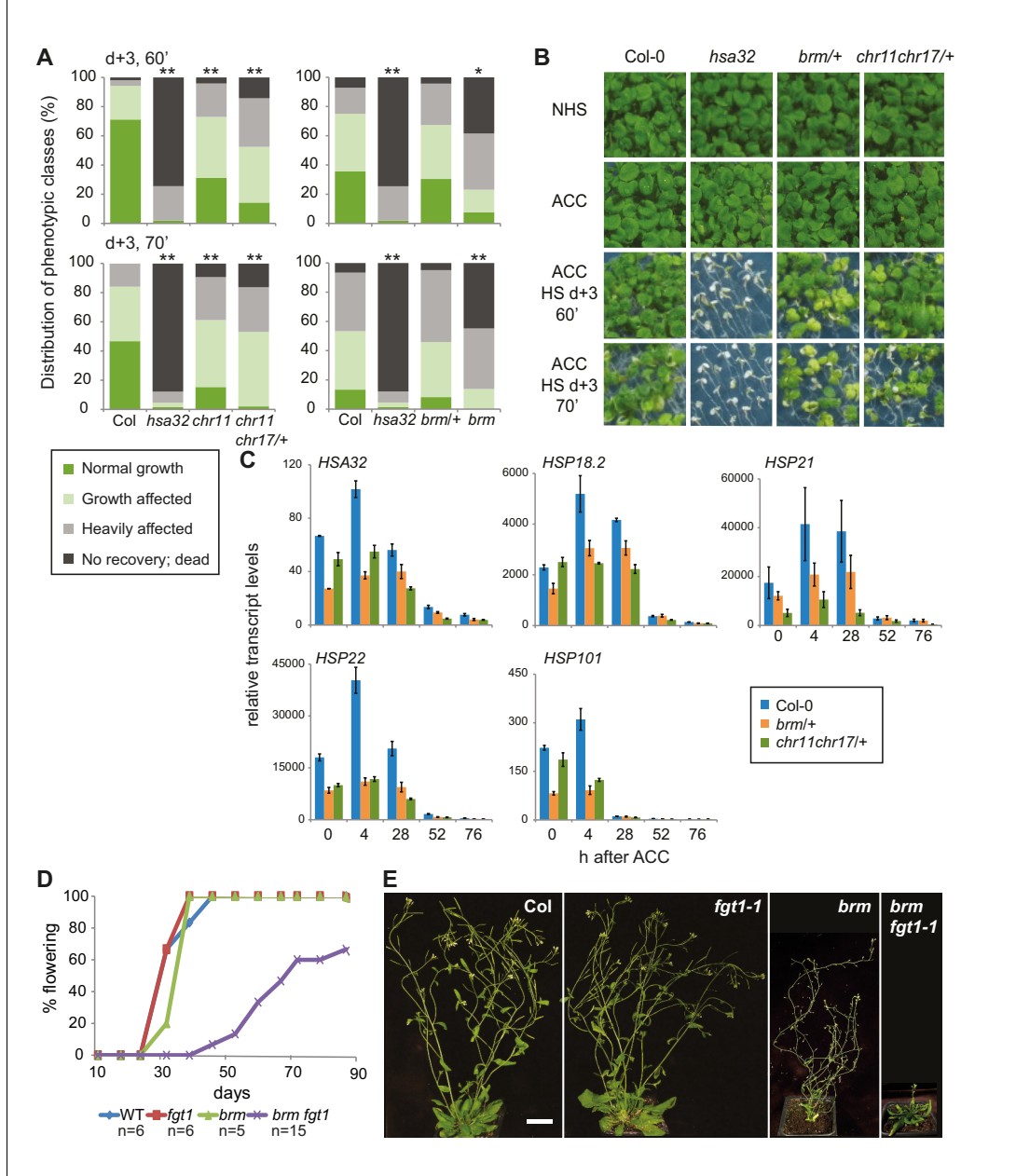

**Figure 6.** The ISWI and BRM chromatin remodelers are required for HS memory and *BRM* interacts genetically with *FGT1*. (**A**) *chr11/chr11*, *chr11/chr11 chr17/+* and *brm-1* mutants show reduced HS memory. Seedlings of the indicated genotypes were acclimatized 5 d after germination and treated with a tester HS 3 d later. *hsa32* was included as a control. Individual seedlings were phenotypically categorized 14 d after ACC and genotyped by PCR. Data are averaged over at least two independent assays, Fisher's exact test, *p<0.05, **p<0.001; n>28. (**B**) Representative picture of the data shown in (**A**) taken 14 d after ACC. (**C**) Transcript levels of HS-inducible genes after ACC are reduced in *brm-1/+* and *chr11/chr11 chr17/+*-segregating lines compared to Col-0. Transcript levels were normalized to *At4g26410* and the corresponding NHS sample. Data are averages and SE of at least three biological replicates. (**D, E**) The *brm-1 fgt1-1* double mutant is delayed in growth and development in long-day conditions. (**D**) Flowering time in days to first open flower. Genotypes were isolated from a segregating population. The percentage of flowering plants is plotted against the days of growth. (**E**) Representative individuals of the indicated genotypes grown for 51 d (Col-0, *fgt1-1*) or 67 d (*brm-1*, *brm-1 fgt1-1*). Size bar, 2 cm.

The following figure supplements are available for figure 6:

**Figure supplement 1.** *CHR11*, *CHR17* and *BRM* are not required for the acquisition of thermotolerance.

**Figure supplement 2.** *CHR11*, *CHR17* and *BRM* are not required for basal thermotolerance.

*Figure 6 continued on next page*

*Figure 6 continued*

**Figure supplement 3.** Seedling phenotype of the *brm-1 fgt1-1* double mutant.The *brm-1 fgt1-1* double mutant displays delayed development at the seedling stage.

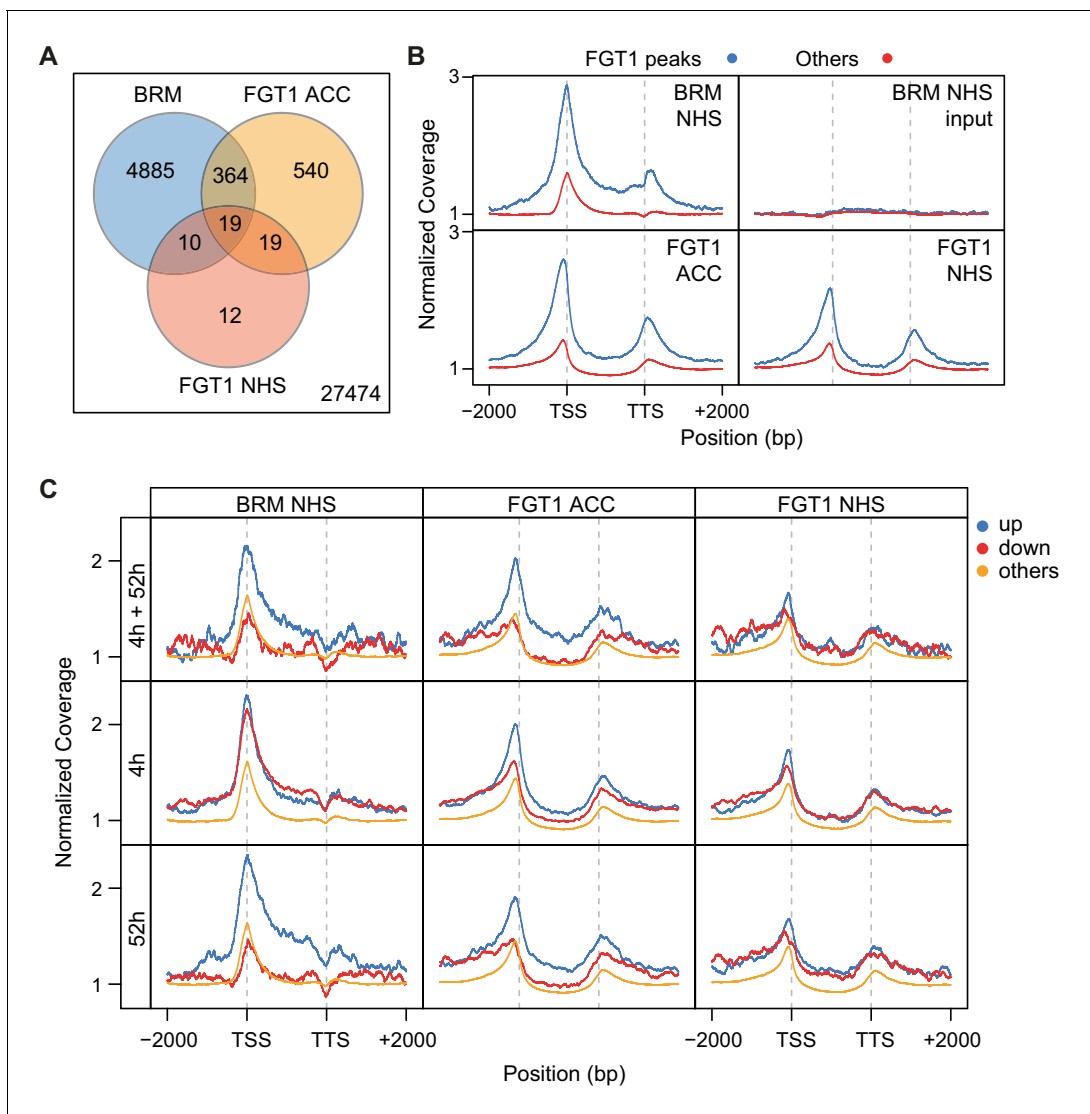

**Figure 7.** BRM and FGT1 show overlapping genomic targeting at HS-responsive genes. (**A**) BRM and FGT1 target genes overlap highly significantly. FGT1 ACC and NHS peak overlapping genes were compared to BRM identified peaks (*Li et al., 2016*). The number of overlapping genes is represented and their significance was estimated by Fisher test (BRM vs. FGT1 NHS $p<10^{-8}$, BRM vs. FGT1 ACC $p<10^{-76}$). (**B**) BRM is enriched at FGT1 target genes and shows a similar coverage profile as FGT1. Normalized coverage profiles of BRM are displayed for FGT1 ACC peak genes and all other genes. FGT1 panels correspond to those in *Figure 4—figure supplement 1*. (**C**) Before HS, BRM binds preferentially to HS memory genes and late HS-induced genes in a pattern similar to FGT1. BRM and FGT1 are strongly enriched at HS memory genes (4 h + 52 h up, top panel, blue line). Normalized read coverages of BRM, FGT1 ACC and FGT1 NHS of genes with changed (up, down, other) expression at 4 and/or 52 h after ACC (*Stief et al., 2014*) are displayed.

The following figure supplement is available for figure 7:

**Figure supplement 1.** Genome browser views of BRM and FGT1 ChIP-seq reads.

## Nucleosome redistribution dynamics after HS are affected in *fgt1-1*

Given the physical and genetic interaction with chromatin remodeling complexes, we next examined the possibility that FGT1 functions by regulating nucleosome dynamics at memory genes. To this end, we determined nucleosome occupancy around the TSS of *HSA32, HSP22.0, HSP18.2* and *HSP101* during two days after ACC in *fgt1-1* by qPCR of Micrococcal Nuclease-digested chromatin (MNase-qPCR). At *HSA32, HSP22.0* and *HSP18.2*, FGT1 was required for the correct positioning and occupancy of the +1 nucleosome already under control conditions (*Figure 8*), consistent with the fact that FGT1 was bound to these loci already before HS (*Figure 3*, *Figure 7—figure supplement 1*). Interestingly, while at *HSA32* occupancy at the +1 nucleosome was reduced in *fgt1-1* compared to parental seedlings, it was increased at *HSP22.0* and *HSP18.2*. In wild type, nucleosome occupancy was strongly reduced at 4 h after ACC for all three genes and recovered slowly over the next two days (*Figure 8—figure supplement 1*). At *HSA32* and *HSP22.0* nucleosome occupancy was still not fully recovered after 52 h. At *HSP18.2* it was fully recovered by 52 h (but not yet 28 h). In *fgt1-1*, nucleosome recovery was accelerated (*Figure 8—figure supplement 1*). Nucleosome dynamics at *HSP101* were not affected in *fgt1-1*. Thus, FGT1 is required to maintain low relative nucleosome occupancy at memory genes after HS.

Because of the sterility and developmental defects of *brm-1* and *chr11 chr17* mutants we were unable to examine nucleosome dynamics in heat-stressed seedings of the remodeler mutants. Given the pre-association of BRM and FGT1 with the memory loci and the findings for *fgt1-1*, we reasoned that nucleosome abundance at these loci may be altered already under non-stress conditions. Indeed, rosette leaves of *fgt1-1*, *brm-1* and *chr11/chr11 chr17/+* showed similarly increased nucleosome abundance under greenhouse conditions for *HSA32, HSP18.2* and *HSP22.0* (*Figure 8—figure supplement 2*). In contrast, the nucleosome abundance at *HSP101* was not changed.

## Discussion

### FGT1 and HS memory

Here, we have reported the identification of the *A. thaliana* orthologue of *Sno*, FGT1, as a regulator of sustained gene induction after HS. We identified FGT1 from an unbiased mutagenesis screen for factors that are required for the maintenance of high *HSA32::LUC* expression after HS, but not for its initial induction. Genome-wide determination of FGT1-binding sites and its high phylogenetic conservation suggest that FGT1 has a general function in promoting gene expression. FGT1 interacts with conserved chromatin remodeling complexes, and the catalytic subunits of these complexes are also required for physiological HS memory. Together, our data suggest the following model (*Figure 9*); HS induces memory gene expression through HSF proteins. FGT1 binds to these loci at the nucleosome-depleted region (NDR) adjacent to the TSS, where it interacts with nucleosome remodeling complexes. By tuning nucleosome occupancy around the TSS, FGT1 maintains these loci in a memory-competent state. This allows active transcription to be maintained for several days after HS, thus contributing to HS memory.

We have previously found that sustained induction and transcriptional memory after HS are associated with HSFA2-dependent H3K4 hyper-methylation (*Lämke et al., 2016*). Whether FGT1 interacts with HSFA2 and H3K4 methylation remains a question for future studies. FGT1 and BRM are associated with HS memory loci already before HS, when their expression levels are low. FGT1 binding increases after HS and is maintained at high levels over the course of HS memory. It is unclear whether HS memory is conserved outside the plant kingdom. However, our results strongly indicate that FGT1 has a more generic role in transcription, despite the lack of morphological aberrations under standard growth conditions. This general function of FGT1 is partially redundant with that of the SWI/SNF remodeler BRM (as apparent in the double mutant). It is conceivable that FGT1 is required for regulated gene expression, which is prevalent especially during development and under stress conditions.

### FGT1 is the *A. thaliana* orthologue of the highly conserved Sno helicase

FGT1 shows a high conservation over the whole protein length with its metazoan orthologues. Sno is a component of the inductive Notch signaling pathway and important for patterning of the *Drosophila* wing margin (*Majumdar et al., 1997*). It is also required for the activation of downstream

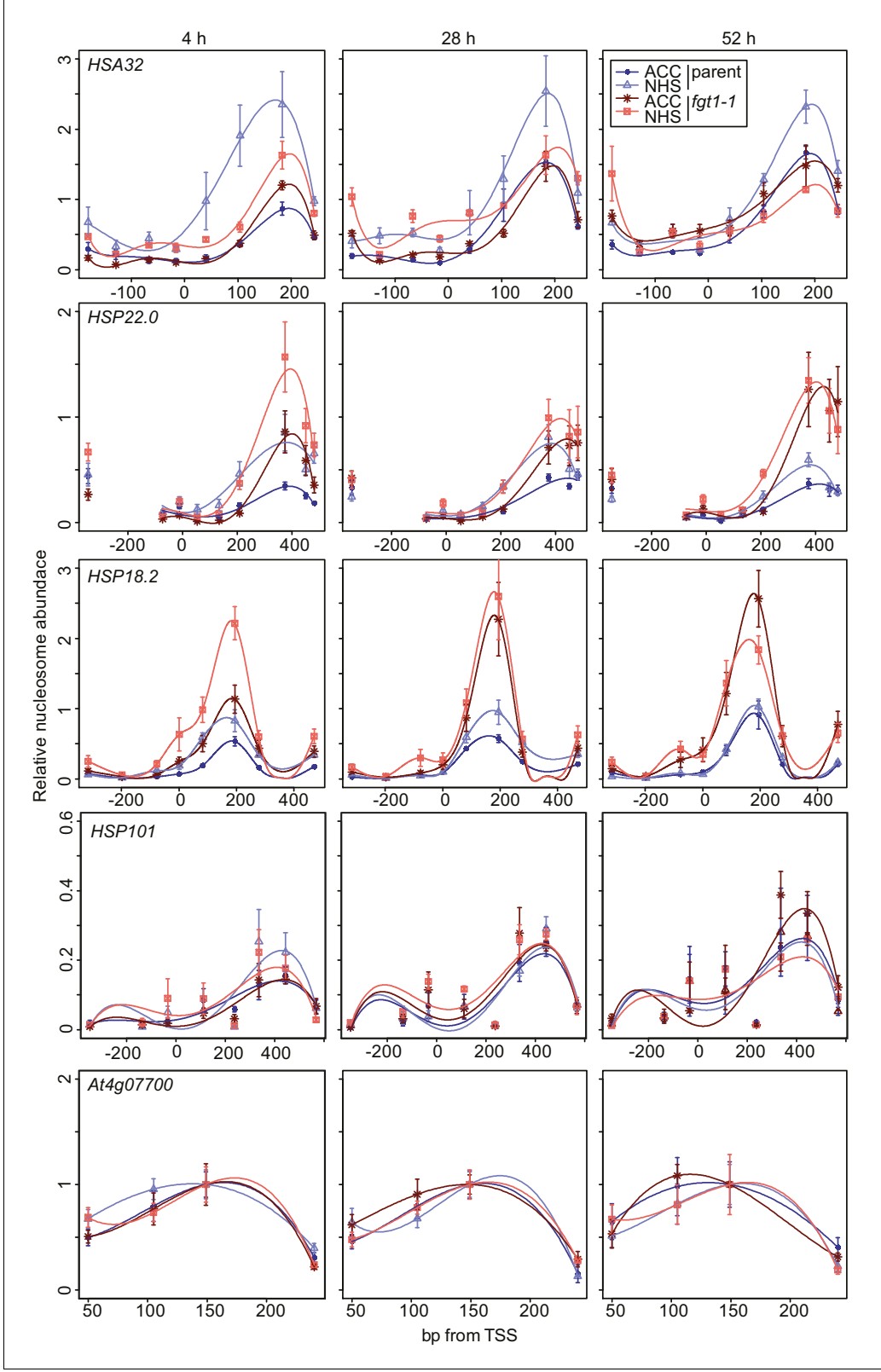

**Figure 8.** *FGT1* is required for nucleosome occupancy and nucleosome recovery after HS at memory genes. *FGT1* is required for proper nucleosome organization around the TSS of memory genes before HS and is required to maintain low nucleosome occupancy during the memory phase. Chromatin dynamics at *HSA32, HSP22.0, HSP18.2, HSP101* and *At4g07700* at 4, 28, or 52 h after acclimatizing HS (ACC) or no HS (NHS) in the parent (blue) or *fgt1-1* (red), respectively. Nucleosome occupancy was determined by MNase-qPCR. Data shown are averages of at least three biological replicates and SE.

*Figure 8 continued on next page*

*Figure 8 continued*

The following figure supplements are available for figure 8:

**Figure supplement 1.** Nucleosome recovery after ACC is delayed in *fgt1-1*.

**Figure supplement 2.** Nucleosome remodeler mutants and *fgt1* show similar nucleosome occupancy defects of memory genes but not *HSP101*.

genes of the Notch and EGFR pathways (*Majumdar et al., 1997*; *Tsuda et al., 2002*). The *C. elegans* orthologue *let-765* was implicated as a positive regulator of lin-3/egf expression in vulval induction (*Simms and Baillie, 2010*), and mammalian SBNO may be involved in neuron development (*Grill et al., 2015*).

FGT1 contains a DExD helicase domain, which is classically considered as an RNA helicase (*Fuller-Pace, 2006*). Interestingly, other DExD/H helicase proteins have been implicated in coordinating transcription and co-transcriptional RNA processing by interacting with co-activator/-repressor proteins (*Fuller-Pace, 2006*). While FGT1 binds to active genes, it was not associated with transcribed regions, but rather with the flanking promoter and terminator sequences, supporting the notion of a role in transcriptional regulation rather than RNA processing. However, we currently cannot rule out a role for (non-)coding RNAs in FGT1-dependent regulation. FGT1 also contains a Helicase C domain, which is frequently found in chromatin remodeler proteins (*Clapier and Cairns, 2009*). In fact, the combination of two helicase domains as present in FGT1 is reminiscent of remodeler proteins. In addition, FGT1 contains a PHD domain. Several PHD domains display very high binding affinities to specific posttranslational modifications of histone H3 (*Musselman and Kutateladze, 2011*). Using recombinant $FGT1_{PHD}$ or transgenic FGT1-YFP, we confirmed H3 binding, however, we did not observe preferential binding to any of the tested modifications (*Figure 2G*, *Figure 2—figure supplement 3*), suggesting that other determinants contribute to targeting FGT1. In agreement with this, the PHD domain of FGT1 could not be assigned to any of the characterized subgroups with high specificity for methylated H3 (*Lee et al., 2009*).

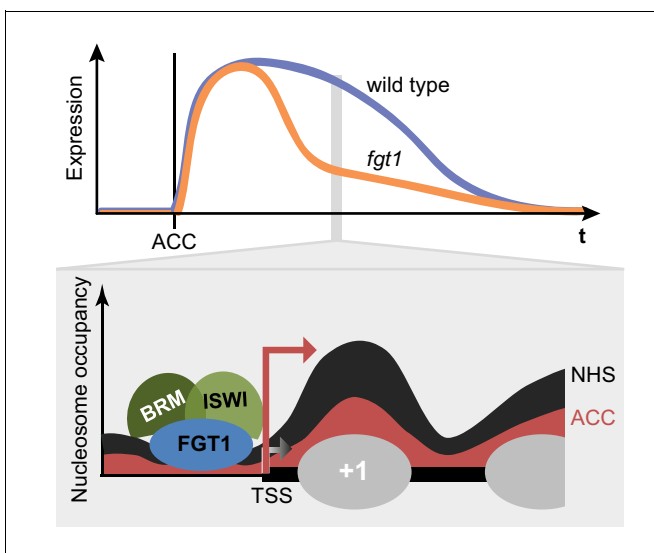

**Figure 9.** FGT1 interacts with chromatin remodelers and affects nucleosome dynamics during transcription and HS memory. Top: Schematic representation of HS memory gene expression in wild type and *fgt1* after acclimatizing HS (ACC). Loss of FGT1 causes loss of sustained gene induction. Bottom: FGT1 interacts with ISWI and BRM remodelers near the TSS to maintain low nucleosome occupancy. In the absence of FGT1, nucleosome recovery is accelerated. Profiles were drawn based on the data obtained for *HSA32*. Angled arrows indicate transcriptional activity.

## FGT1 interacts with chromatin remodeling proteins

Chromatin remodeling complexes use the energy of ATP to move nucleosomes along DNA, and to eject or exchange nucleosomes on DNA (*Clapier and Cairns, 2009*; *Narlikar et al., 2013*; *Struhl and Segal, 2013*), thus changing the accessibility of DNA for other proteins. Chromatin remodeling complexes have widespread functions during development and have been linked with several pathologies, including tumorigenesis (*Narlikar et al., 2013*). Four families of chromatin remodelers are conserved in metazoans, plants and yeast; the SWI/SNF, the ISWI, the CHD and the INO80 families (*Clapier and Cairns, 2009*). The catalytic subunits of all families contain a Snf2-related helicase domain and form large multi-subunit complexes. Although they have differing functions, there is evidence that remodelers of different families interact on the chromatin (*Clapier and Cairns, 2009*; *Narlikar et al., 2013*). FGT1 interacts with remodelers of the ISWI and SWI/SNF families. Both ISWI orthologues, BRM and several accessory subunits (SWI3, BAF73b) were identified by co-immunoprecipitation. ISWI is required for the regular spacing of nucleosomes downstream of the +1 border nucleosome (*Clapier and Cairns, 2009*; *Li et al., 2014*). FGT1 was found previously in a co-immunoprecipitation experiment with CHR17, thus independently confirming our results (*Smaczniak et al., 2012*). Drosophila BRM was originally identified due to its activating function that antagonizes Polycomb group silencing (*Tamkun et al., 1992*; *Clapier and Cairns, 2009*). During HS memory, both BRM and ISWI act as positive regulators of the memory, suggesting that both contribute to maintaining the chromatin open and accessible. FGT1, localized just upstream of the TSS, may bridge both remodelers. This notion is consistent with the localization of BRM and ISWI in other systems (*Gkikopoulos et al., 2011*; *Yen et al., 2012*; *Narlikar et al., 2013*) and with the high overlap in BRM and FGT1 localization (*Figure 7* and *Li et al., 2016*). Moreover, remodelers of different families cooperate to regulate dynamic sites (*Morris et al., 2014*). The evidence for interaction with BRM is further strengthened by the double mutant analysis, which demonstrated that *FGT1* activity becomes critical in the absence of *BRM*.

## FGT1 modulates nucleosome occupancy near the TSS

FGT1 binds to expressed genes just upstream of the TSS and just downstream of the TTS. Whether this reflects one function or two separable functions, remains to be investigated. The FGT1 peaks that we detected were wide and relatively flat. Consistent with FGT1 interacting with BRM and CHR11/CHR17, such a binding pattern is characteristic for chromatin remodeling proteins but not sequence-specific transcription factors (*Gkikopoulos et al., 2011*; *Yen et al., 2012*; *Zentner and Henikoff, 2013*; *Li et al., 2016*). FGT1 was preferentially associated with medium and highly expressed genes. Among the lowly expressed genes, those that are HS-inducible were more strongly bound than other genes - not only after HS, but also under control conditions. This indicates that despite an overall correlation with expression, FGT1 binding does not simply reflect the transcriptional activity of a locus. It is tempting to speculate that at some loci FGT1 binding indicates a readiness for transcription rather than actual transcription, similar to what was described as a poised state (*Levine et al., 2014*). FGT1 binding overlaps the NDRs present next to the TSS and TTS. Hence, FGT1 may be required to maintain the NDR by interacting with remodeler proteins. Accordingly, we observed changes in the nucleosome occupancy of this region and particularly the adjacent +1 nucleosome in *fgt1-1* mutants. The changes in nucleosome occupancy at FGT1 target loci preceded HS, in agreement with the FGT1 binding pattern. In non-stressed adult leaves, similar changes were observed for *brm* and *iswi* mutants. Locus-specific and developmental effects indicate the involvement of additional components. Thus, it is likely that BRM and ISWI cooperate with FGT1 to mediate nucleosome occupancy before and after HS. We propose that the molecular function of Sno/FGT1 orthologues is conserved and that they promote transcriptional regulation through interaction with chromatin remodeling complexes. The reported function of *Sno* as a co-activator is consistent with this mechanism; moreover, the interacting chromatin remodelers are highly conserved in metazoans.

In summary, we have uncovered a role for nucleosome occupancy in stress memory that is modulated by the conserved chromatin regulator FGT1. Sno/FGT1 also functions in a broader context to sustain gene expression during development, stress adaptation and pathologies. Our results identify a mechanism of how environmentally-induced gene expression is sustained after cessation of an

external cue and provides a molecular framework for a chromatin memory. This mechanism may be exploited to improve stress tolerance in crop plants.

## Materials and methods

### Plant materials, growth conditions and HS assays

*Arabidopsis thaliana* Col-0 seedlings were grown on GM medium (1% (w/v) glucose) under a 16 h/ 8 h light/dark cycle at 23°C/21°C. *brm-1*, *hsa32-1* and *hsp101* were previously described (*Charng et al., 2006*; *Hurtado et al., 2006*; *Stief et al., 2014*). *chr11-1$^{-/-}$chr17-1$^{+/-}$* was obtained from K. Kaufmann (*Li et al., 2012*; *Smaczniak et al., 2012*). T-DNA insertion lines in *FGT1* (*fgt1-2*, SALKseq_17372; *fgt1-3*, SALK_036520) and *brm-1* were obtained from NASC. Heat treatments were performed on 4 d-old seedlings unless stated otherwise. Seedlings were treated with an acclimatizing HS (ACC) of 37°C for 60 min, followed by 90 min at 23°C and 45 min at 44°C starting eight hours after light onset. As tester HS a 44°C treatment for the indicated times was applied. After HS, plants were returned to normal growth conditions. Thermotolerance assays were performed as described (*Stief et al., 2014*). For all assays, all genotypes of one treatment were grown on the same plate.

### Construction of transgenic lines

To generate *HSA32::HSA32-LUC*, a 4.6 kb fragment encompassing the complete *HSA32* gene and 2.2 kb of promoter sequences was amplified (pHSA32_for_SacI GTGGAGAGCTCAAAGCTGCCA TGAATGTGTT, HSA32_rev_PstI AACACTGCAGACAATGCCAAGTTTGATGCCTGA) from genomic DNA, the Stop codon was mutated and replaced by the LUC reporter gene. The resulting *HSA32:: HSA32-LUC* construct was transformed into Col-0. For *pFGT1::FGT1* (SA13) a 10.7 kb genomic fragment encompassing the complete *FGT1* gene and including 1.5 kb promoter sequences was amplified (EMBFrag1_FSphI TACTGCATGCCTTTAGCGTTATCGAATCT, EMBFrag4_RBamhI CAAGAGG TTAGGATCCGCTTCCAGACA) and inserted into pBarMAP (ML516). To make *35S::FGT1-YFP* (SA1), *FGT1* was amplified from cDNA mutating the stop codon (EMB1135BamHI_F AGGGATCCACAA TGACGCAGTCGCCTGTTCAAC, EMBBamHInoStpR A AGG ATC CGC ATC ATC AAT CTC TTG AAC CCA TGC T) and inserted into *pBarM:35S::YFP* (IB30), thus generating *pBarM:35S::FGT1-YFP*. For *pFGT1::FGT1-YFP* (EB19) the *FGT1* stop codon in SA13 was mutated to a *Sal*I site into which YFP was inserted to generate *pFGT1::FGT1-YFP*. All constructs were inserted into *Agrobacterium tumefaciens* strain GV3101 and transformed into *A. thaliana* using the floral dip method (*Clough and Bent, 1998*).

### Reporter gene analyses

LUC activity was detected by spraying seedlings with 2 mM Luciferin (Promega, Mannheim, Germany) and imaging with a Nightowl (Berthold Technologies, Bad Wildbad, Germany). Image analysis was performed with IndiGO software (Berthold). The signal threshold was adjusted to the signal of the parental line. YFP and RFP fluorescence was imaged using a Zeiss LSM710 confocal microscope (Zeiss, Jena, Germany). Controls were imaged using the same settings and laser intensities. For BiFC YFPN-FGT1, YFPC-CHR11, YFPC-CHR17 and YFPC-BRM constructs were generated using published vectors pE-SPYCE and pE-SPYNE, respectively (*Walter et al., 2004*; *Weltmeier et al., 2006*). Combinations of YFPN and YFPC fusion constructs were co-expressed in four to six week-old *Nicotiana benthamiana* leaves using leaf infiltration of *A. tumefaciens* GV3101 suspensions containing the tested construct combinations. Fluorescence in the YFP spectrum was analysed 2 d after infiltration. A *UBC10::BRU1-RFP* construct was co-transformed to image nuclei (*Ohno et al., 2011*).

### Gene expression analysis

RNA extraction, reverse transcription and qPCR were performed as described previously (*Stief et al., 2014*; *Lämke et al., 2016*). *At4g26410* was used as a reference gene (*Czechowski et al., 2005*). Primer sequences are listed in *Supplementary file 1*.

### Histone peptide binding assays

The PHD domains of FGT1 (aa 667–757) and ING1 (*Lee et al., 2009*) were subcloned into pGEX-4T-1 (Amersham Biosciences, Pittsburgh, PA), expressed in *E. coli* BL21 (DE3) as GST fusion proteins

and purified using Glutathione affinity resins (Thermo Fisher Scientific, Waltham, MA). The histone peptide binding assay was performed as described (*Kabelitz et al., 2016*). In brief, 1 µg of biotinylated histone peptides (Merck-Millipore, Darmstadt, Germany) were incubated with 10 µg of GST-fusion protein in binding buffer (50 mM Tris-HCl pH 7.5, 300 mM NaCl, 0.1% NP-40, 1 mM PMSF, protease inhibitors) overnight at 4°C with rotation. After incubation with Streptavidin Dynabeads (Thermo Fisher Scientific) and extensive washing with TBST, bound proteins were analyzed by SDS-PAGE and immunoblotting with anti-GST antibodies (Merck-Millipore). Results for GST-ING1 were taken from *Kabelitz et al. (2016)*, as the experiments were performed in parallel.

## ChIP-qPCR, MNase-qPCR

ChIP was performed as described (*Lämke et al., 2016*). MNase-qPCR was performed as described (*Liu et al., 2014*). Primer sequences are listed in *Supplementary file 1*. Curves were created based on a polynomial regression (*HSA32, HSP22.0, HSP101, At4g07700*) or spline interpolation (*HSP18.2*). The position of the center of each amplicon relative to the TSS is indicated. The +1 nucleosome of *At4g07700* was used for normalization (*Kumar and Wigge, 2010*).

## ChIP-seq analysis

ChIP-seq was done with three biological replicates for two genotypes (*35S::FGT1-YFP*, Col-0) and two treatments (acclimated (ACC) and control (NHS)). 5 d-old seedlings were treated with ACC or NHS and harvested 28 h later. ChIP was performed as described above. Library preparation and 100 bp single-end sequencing on an Illumina HiSeq 2500 were performed by ATLAS Biolabs (Berlin, Germany). Data were delivered as de-multiplexed fastq files. Raw data have been deposited at NCBI SRA under accession number SRA GSE79453. BRM ChIP-seq raw data (*Li et al., 2016*) were downloaded from NCBI SRA. All statistical analyses were done using R (http://www.r-project.org). Figures were made using R base plotting or the lattice package.

Read adapters were removed using Trimmomatic (*Bolger et al., 2014*). The resulting reads were mapped against the *A. thaliana* TAIR10 reference genome using bwa mem (*Li, 2013*). Mapping files were further transformed into sorted bam files and indexed using samtools (*Li et al., 2009*). Duplicate reads were removed using samtools rmdup. One Col-0 ACC sample was filtered out due to its very low number of mapped reads.

Peak calling was done using MACS (*Zhang et al., 2008*) for each sample separately, lower mfold was decreased to 2. Only peaks called for all three FGT1 samples with the same treatment and not called for any of the 2 Col-0 control samples were considered as a signal. Distances between the peaks thus identified and annotated genes were calculated using bedtools closestBed (*Quinlan and Hall, 2010*). The expression of those genes was estimated based on published RNA-seq data for 11 d-old seedlings (*Gan et al., 2011*).

For coverage profiles, base coverages for the whole genome and selected regions were computed using bedtools coverage and normalized by division by the total number of covered bases (as calculated by multiplying the number of mapped reads by their length), chloroplast and mitochondrion sequences were excluded. Coverage profiles around genes were made using 2 kb prior to the TSS, gene regions, and 2 kb after the TTS. Values were averaged over biological replicates. Genic (transcribed) region base coverages were scaled to the same length to make them comparable between different genes. Genes were categorized into not expressed genes and equally sized groups of highly, moderately and lowly expressed genes according to their expression under standard conditions as determined by a published RNA-seq experiment (*Gan et al., 2011*). This dataset corresponds to the closest developmental stage and environmental conditions regarding our ChIP-seq experiment with available data at NCBI GEO (GSM764077). Coverage profiles for each class were calculated by averaging the values for all member genes for each genotype x treatment combination. Another classification was done by grouping genes based on their expression pattern 4h after ACC as determined by ATH1 microarray hybridization (*Stief et al., 2014*). Coverage profiles for those classes were calculated as described above.

FGT1 binding enrichment depending on chromatin state was investigated by comparing coverages for chromatin states as described (*Sequeira-Mendes et al., 2014*). Coverages were normalized regarding the total number of covered bases and the length of each chromatin state region. Values for each state were averaged at the sample level.

## Immunoprecipitation and mass spectrometry

Native FGT1-YFP protein complexes were immunoprecipitated from *35S::FGT1-YFP* seedlings and subjected to nHPLC-MS/MS for identification. In detail, 2.5 g of 4 d-old seedlings subjected to ACC or NHS were harvested 28 h later and snap-frozen in liquid nitrogen. Nuclei were extracted according to (*Kaufmann et al., 2010*) and sonified using a Diagenode Bioruptor (3 cycles/ 30 s on/off) on low intensity settings. Protein extracts were incubated with α-GFP paramagnetic beads for 1.5 h at 4°C and native protein complexes recovered using α-GFP isolation kit (Miltenyi Biotec, Bergisch Gladbach, Germany) and eluted in 8 M urea (Sigma-Aldrich, München, Germany). Eluates were diluted and digested with trypsin (Promega) as described (*Smaczniak et al., 2012*). Peptides were desalted, lyophilized and re-suspended in 30 µL 5% (v/v) acetonitrile, 2% (v/v) trifluoroacetic acid. Measurements were performed on a Q Exactive Plus orbitrap mass spectrometer coupled with an Easy nLC1000 HPLC (Thermo Fisher Scientific). Spectra were analyzed using MaxQuant software (*Cox and Mann, 2008*) and the *A. thaliana* TAIR10 annotations. A decoy database search was used to limit false discovery rates to <1% on the protein level.

## Acknowledgements

We thank the European Arabidopsis stock centre and K Kaufmann for seeds. We thank K Henneberger, V Schüler, K Sklodowski for technical assistance, members of our laboratory, A Sicard and M Lenhard for helpful comments. We thank C Schmidt and D Mäker for excellent plant care.

## Additional information

### Funding

| Funder | Grant reference number | Author |
| --- | --- | --- |
| Alexander von Humboldt-Stiftung | Sofja-Kovalevskaja-Award | Isabel Bäurle |
| Deutsche Forschungsgemeinschaft | SFB973, Project A2 | Isabel Bäurle |
| Royal Society | University Resarch Fellowship | Isabel Bäurle |

The funders had no role in study design, data collection and interpretation, or the decision to submit the work for publication.

### Author contributions

KB, SA, IB, Conception and design, Acquisition of data, Analysis and interpretation of data, Drafting or revising the article; HC, PN, EB, TK, AG, Acquisition of data, Analysis and interpretation of data, Drafting or revising the article; MG, FJ, Acquisition of data, Analysis and interpretation of data; CK, Conception and design, Analysis and interpretation of data, Drafting or revising the article

### Author ORCIDs

Michal Gorka, http://orcid.org/0000-0002-1289-6858
Christian Kappel, http://orcid.org/0000-0002-1450-1864
Isabel Bäurle, http://orcid.org/0000-0001-5633-8068

## Additional files

### Supplementary files

• Supplementary file 1. Sequences of oligonucleotides used in this study.

### Major datasets

The following dataset was generated:

| Author(s) | Year | Dataset title | Dataset URL | Database, license, and accessibility information |
|---|---|---|---|---|
| Brzezinka K, Altmann S, Czesnick H, Nicolas P, Benke E, Kabelitz T, Kappel C, Bäurle I | 2016 | Arabidopsis FORGETTER1 sustains stress-induced transcription through nucleosome remodeling | http://www.ncbi.nlm.nih.gov/geo/query/acc.cgi?acc=GSE79453 | Publicly available at NCBI Gene Expression Omnibus (accession no: GSE79453) |

The following previously published datasets were used:

| Author(s) | Year | Dataset title | Dataset URL | Database, license, and accessibility information |
|---|---|---|---|---|
| Richard M Clark | 2011 | Multiple reference genomes and transcriptomes for Arabidopsis thaliana | http://www.ncbi.nlm.nih.gov/geo/query/acc.cgi?acc=GSE30814 | Publicly available at NCBI Gene Expression Omnibus (accession no: GSE30814) |
| Li C | 2016 | Genome-wide profiling of SWI/SNF chromatin remodeler BRAHMA and Histone H3 lysine demethyalse Relative of Early Flowering 6 (REF6) in Arabidopsis | http://www.ncbi.nlm.nih.gov/sra/SRR2243593 | Publicly available at NCBI Sequence Read Archive (accession no: SRR2243593) |
| Li C | 2016 | Genome-wide profiling of SWI/SNF chromatin remodeler BRAHMA and Histone H3 lysine demethyalse Relative of Early Flowering 6 (REF6) in Arabidopsis | http://www.ncbi.nlm.nih.gov/sra/SRR2243594 | Publicly available at NCBI Sequence Read Archive (accession no: SRR2243594) |

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
