## [Decision Letter]

Thank you for submitting your article "*Arabidopsis* FORGETTER1 mediates stress-induced chromatin memory through nucleosome remodeling" for consideration by *eLife*. Your article has been reviewed by three peer reviewers, and the evaluation has been overseen by Detlef Weigel as the Senior and Reviewing Editor. The following individual involved in review of your submission has agreed to reveal his identity: Yijun Qi (Reviewer #1).

The reviewers have discussed the reviews with one another and the Reviewing Editor has drafted this decision to help you prepare a revised submission.

Summary:

Through a mutant screen, you have identified a new gene, FORGETTER1 (FGT1), that is required for maintenance of heat-induced gene expression. You propose that the FGT1 protein globally associates with the promoter regions of actively expressed genes in a heat-dependent fashion, interacting in this process with chromatin remodelers of the SWI/SNF and ISWI families. You conclude that FGT1 mediates stress-dependent chromatin memory by modulating nucleosome occupancy.

The reviewers agreed that such findings should in principle be published at the highest levels. While the reviewers furthermore agreed that the data presented are clear, they also agreed that the conclusions went considerably beyond what the data showed directly. To bring the interpretation more in line with the data, they ask you to tone down your claims, and also to add a series of new data to support their claims:

Essential revisions:

1) Perform a carefully controlled RNA-seq time course experiment (wt, fgt1-1, brm1 etc) over a heat shock to compare fgt1 and swi/snf effects more precisely. Alternatively, test whether SWI/SNF and ISWI is required for FGT1 binding to HS memory genes or vice versa, with FGT1 ChIP in the swi/snf or iswi mutants and SWI/SNF and ISWI ChIP in the fgt1 mutant, and add RNAPII ChIP.

2) Examine whether nucleosome redistribution dynamics during the HS memory phase are affected by swi/snf or iswi mutations.

3) Include as a control analysis of nucleosome occupancy of a gene that is not a target of FGT1 nor persistently expressed (i.e. HSP101).

4) Provide a better control for FGT1-H3 interaction.

We have included the full reviews, which mention separate points for essential revisions, but after discussion, we condensed them to the four points above

Reviewer #1:

In this manuscript, the authors identified FORGETTER1 (FGT1) as a new factor that is required for heat stress (HS) memory. They found that FGT1 primarily binds to the proximal promoter upstream of the TSS as well as the region downstream of the TTS and functions as a co-activator of memory genes. Moreover, they showed that FGT1 interacts with SWI/SNF and ISWI chromatin remodelers that are also required for HS memory. Finally, they showed that FGT1 is required for maintaining low nucleosome occupancy during the memory phase. The authors propose that FGT1 modulates HS memory through modulating nucleosome occupancy dynamics and transcription-competent chromatin. These findings advance our understanding about the molecular mechanism of stress memory and should be of general interest. The data are of high quality and well presented. Meanwhile, I feel the authors should make efforts to provide the missing link between FGT1 and nucleosome dynamics. My requests for additional experiments are listed below.

1) The interaction between FGT1 and SWI/SNF and ISWI was shown by IP Mass-spec and BiFC experiments. As this is one of the key points of this paper, additional approaches should be employed to demonstrate the interaction.

2) The authors should test whether SWI/SNF and ISWI is required for FGT1 binding to HS memory genes or vice versa. This can be done by performing FGT1 ChIP analyses in the swi/snf or iswi mutants and SWI/SNF and ISWI ChIP in the fgt1 mutant.

3) The authors should examine whether nucleosome redistribution dynamics during the HS memory phase are affected by swi/snf or iswi mutations.

4) It would be also interesting to examine which histone marks are affected by fgt1 during HS memory.

Reviewer #2:

Organisms respond to changes in environmental conditions through changes in transcription. Most of these changes are transient and are lost if the environmental stimulus is removed. For example, heat shock leads to transient upregulation of protein folding chaperones. However, in Arabidopsis, heat stress also leads to persistent expression of several genes (HSA32, HSP21, HSP22.0, HPS18.2) for ~3 days and this protects plants from lethal heat stress during this period. This phenomenon is called heat stress memory and requires an HSF-related transcription factor HSFA2 and is correlated with changes in histone H3 lysine 4 methylation.

Here, the authors perform a genetic screen for mutants that fail to express HSA32 three days after heat stress. This screen identified a mutant in a gene called FORGETTER (FGT1), which is homologous to Strawberry Notch in animals. This mutant resulted in 1) normal expression of HSA32 immediately after a mild heat shock, 2) normal survival of a lethal heat stress administered 90 minutes after a mild heat stress, 3) low expression of an HSA32 reporter (compared with wild type) after 2 or 3 days of recovery and 4) poor survival of a lethal heat stress after 3 days of recovery. These results suggest that FGT1 is specifically required for the long-term effects of mild heat stress. The manuscript seeks to understand the phenotype of the fgt1 mutant and to define the molecular role of FGT1 in stress memory. This is a very interesting phenomenon and the system is excellent. The experiments are well designed and the data are clear. However, I do not agree with the authors' interpretation of several experiments and I am not convinced that they support the major conclusions of the paper. As a result, I do not feel that this work is mature enough to recommend publication in *eLife*.

To validate these observations, the authors measured the mRNA levels of both genes that exhibit persistent expression after heat stress (HSA32, HSP21, HSP22.0 and HSP18.2) and a gene that is induced by heat stress but does not show persistent expression (HSP101) in the fgt1-1 mutant. Surprisingly, despite having used an HSA32 reporter for the screen, the fgt1-1 mutant plants showed normal persistent expression of HSA32 and HSP18.2 and slightly defective persistence of HSP21 and HSP22.0. However, the authors claim that the very slight decrease in HSA32 after 45 and 69h are the source of the phenotype. Because there was no statistical analysis of this difference, it is unclear if it is significant. These differences do not appear to be as strong as the differences observed with the Luciferase reporter (Figure 1). Quantitation of the luciferase for comparison would have clarified the phenotype. To convince the reader that the mutation is impacting transcription, ChIP against RNAPII would have helped. Furthermore, inclusion of the HSFA2 mutant for comparison would have made the strength of the phenotype easier to assess. It is possible that the mutation has a larger effect on the reporter and that the survival phenotype is the result of an effect on a different subset of genes (i.e., HSP21 and HSP22). If so, then the persistent expression of genes in response to heat stress involves both an FGT1-dependent and an FGT1-independent mechanism.

The authors hypothesize that FGT1 promotes persistent expression of HSP21 and HSP22.0 by remodeling nucleosomes because the protein possesses a PHD finger domain and a Helicase C domain. Consistent with this hypothesis, FGT1 localizes to the nucleus, interacts with both active genes and heat stress-induced genes and physically interacts with the BRM (SWI/SNF) and CHR11/CHR17 (ISWI). Unfortunately, a number of experiments create uncertainty about this conclusion. First, the changes in nucleosome occupancy in the fgt1 mutant do not correlate with binding of FGT1: FGT1 binds strongly to the HSA32 promoter/TSS upon heat stress and this correlates with decreased nucleosome occupancy. The fgt1 mutant shows lower nucleosome occupancy prior to heat stress and this decreases to a comparable level to the wild type in the days following heat stress. Therefore, FGT1 is not required for the decrease in nucleosome occupancy at HSA32. In the cases of HSP18.2 and HSP22, although the fgt1 mutant shows increased nucleosome occupancy, the changes in the wild type strain upon heat stress are difficult to appreciate and, as mentioned above, there is no obvious effect of this mutation on the expression of HSP18.2. It would have been helpful to include a gene that is neither a target of FGT1 nor persistently expressed (i.e. HSP101) as a more relevant control. Likewise, as mentioned above, examining the effect of the HSFA2 mutant in this assay would have been informative. Also, peptide binding experiments show that the PHD finger interacts non-specifically with (non-overlapping) histone peptides. The authors did not include a biologically irrelevant peptide control, but the low affinity and the lack of specificity cast doubt on the claim that "FGT1 binds to the N-terminal region of H3[…]". It seems equally plausible that FGT1 binds to the FLAG peptide. Finally, ChIP seq shows that FGT1 associates with both the transcription start sites and the transcription termination sites of active genes. Given the DExD-like helicase domain, this raises the alternative hypothesis that the protein is involved in post-transcriptional events and might impact the ability of plants to survive through influencing protein levels.

The authors also explored the role of BRM and CHR11/CHR17 in heat stress memory. Mutations in BRM1 and FGT1 produced synthetic flowering and development phenotypes, suggesting non-overlapping roles in these processes. Both mutant plants were somewhat defective in heat stress protection after 3d of recovery. However, these mutant plants also showed significantly lower peak levels of the HS-induction of HSA32, HSP18.1, HSP21 and HSP22. Therefore, the decrease in expression afterward appears proportional to the wildtype, but starting at a significantly lower peak level. If so, then poor survival after 3d of recovery is likely due to lower levels of induced transcription, not less sustained transcription.

Reviewer #3:

How plants adapt to heat stress and remember previous heat shocks is a very interesting area of biology. This paper breaks new ground by discovering a new component, FORGETTER1, which the authors show is necessary for plants to "remember" heat stress correctly at the level of transcriptional regulation. Interestingly, the fgt1-1 mutants have a specific defect in acquired thermotolerance, consistent with the proposed role for this gene in controlling the transcriptional response to temperature. The authors present a rather complete story, since they go from a genetic screen, to mapping and examining the underlying molecular mechanism. It is a bit of an open question as to the actual role of FGT1 in transcriptional memory itself, but that is probably beyond the scope of this study.

Here are a few suggestions that came to my mind on reading the paper, and resolving these might make the study even stronger:

1) For the ChIP-seq of FGT1 it would be nice to see screen shots of actual loci from a browser. Averaged data is rather hard to interpret. Relating to this, different gene lists were used for "active" genes and HS responsive genes for Figure 4. Isn't there a more relevant list of genes whose transcription appears to show "memory" of heatshock? Another way of looking at this, what about the transcriptome of fgt1 itself?

2) The model for FGT1 action is interesting. It seems to invoke a specific role for +1 nucleosomes in responding to temperature. It's striking that there is such a strong enrichment for FGT1 at the +1 position as well as the TTS. Of course these are sites where there is global enrichment for H2A.Z-Nucleosomes (Coleman-Derr and Zilberman 2012) and H2A.Z-nucleosomes are evicted in response to temperature (Kumar and Wigge 2010). It is therefore curious that no attempt is made to synthesise what is already known about temperature-dependent gene expression and its regulation by chromatin. At a minimum, one could simply overlay the distribution of FGT1 with that of H2A.Z.

---

## [Author Response]

*Essential revisions:*

*1) Perform a carefully controlled RNA-seq time course experiment (wt, fgt1-1, brm1 etc) over a heat shock to compare fgt1 and swi/snf effects more precisely. Alternatively, test whether SWI/SNF and ISWI is required for FGT1 binding to HS memory genes or vice versa, with FGT1 ChIP in the swi/snf or iswi mutants and SWI/SNF and ISWI ChIP in the fgt1 mutant, and add RNAPII ChIP.*

This revision concerns the interaction of FGT1 and the SWI/SNF and ISWI chromatin remodelers globally and the level at which FGT1 acts. We have addressed this issue via two complementary approaches. (1) We have analyzed the overlap in genome-wide binding sites of the SWI/SNF protein BRM with those of FGT1. The BRM ChIP-seq dataset employed for this purspose has recently been published (Li et al., Nat Genet 2016), using a BRM-GFP line and anti-GFP antibody in non-stressed conditions. At the genome-wide level, we detected a highly significant overlap between regions bound by BRM and FGT1 as shown in Figure 7. In particular, BRM associates significantly more strongly with FGT1-bound regions than with the rest of the genome and both factors show a very similar binding profile across genes (Figure 7). The same enrichment of BRM binding is also seen for genes that show sustained expression after HS in our time-course microarray experiments (Figure 7, Stief et al. 2014). These results are supported by a series of browser screenshots of individual genes, as suggested by Reviewer #3 (Figure 7—figure supplement 1). Thus, we conclude that a high fraction of FGT1-bound regions is also bound by BRM and the significant overlap in these orthogonal genome-wide datasets strongly supports our interpretation of a functional interaction between the two proteins.

(2) To strengthen the conclusion that FGT1 acts at the transcriptional level, we have added the analysis of unspliced transcripts of one additional memory gene (HSP21) and two additional HS-inducible non-memory genes (HSP70, 101), which clearly show that the effect is specific for memory genes and occurs before splicing, i. e. (co-) transcriptionally (Figure 1—figure supplement 3). The other memory genes do not contain introns, thus precluding this type of analysis. Moreover, the transcript analysis of the LUC transgene shows that the magnitude of the effect is similar to that observed for the endogenous HSA32 gene, suggesting that the transgene mimicks closely the behavior of the endogeneous HSA32. This is in accordance with the analysis of unspliced HSA32 transcripts (Figure 1), for which the primers detected both the endogene and the transgene. To clarify the significance of the differences, we have also added statistical analysis for the qRT-PCR experiments presented in Figure 1 and Figure 1—figure supplement 3. Regarding the RNA pol II ChIP, we have not been able to obtain reproducible chromatin immunoprecipitation with the available antibody (abcam ab817) despite several attempts by different researchers in the group with different antibody lots. Together, these two approaches support the functional interaction of BRM and FGT1 at the transcriptional start sites of a subset of genes genome-wide.

Besides providing this additional experimental evidence, we have toned down the relevant conclusions, bringing them closely in line with what we have directly shown experimentally and we have adjusted the model in Figure 9. For example, we have rephrased where previous wording suggested that FGT1 is recruiting the remodelers (see Discussion).

Regarding the additional suggestions, we feel that the additional qRT-PCR expression analysis (see above) and the strong overlap of BRM and FGT1 binding alleviate the immediate need for the extensive RNA-seq experiment to a large degree. Together with the phenotypic similarity in the memory assays (Figure 6) and the synergistic double mutant phenotypes, the above overlap analysis of genomic binding provides clear support for our interpretation of a functional interaction between FGT1 and SWI/SNF chromatin remodelers at the genome-wide level; this is particularly convincing to us, as it is based on completely independent datasets.

As for the additional ChIP experiments, these are unfortunately not feasible at the moment. We have not been able to obtain sufficient quantities of anti-BRM antibody (Archacki et al., Planta 2009) to perform a solid analysis of BRM binding in wild type and fgt1 mutants. The preliminary results suggest that FGT1 is not required for binding of BRM to memory genes.

Both the brm-1 and the chr11/17 double mutants are sterile (Hurtado et al. 2006; Li et al. 2012), so after crossing in the FGT1-YFP transgene it would not be possible to obtain sufficient homogeneous starting material for these ChIP experiments at the relevant seedling stage, as our protocol requires two grams of starting material. The brm-1 homozygous mutants cannot be distinguished with certainty from wild-type seedlings up until about 12 days after germination, i.e. much later than when we perform all our analyses; the chr11/17 double mutant, by contrast, is severely dwarfed and occurs at a very reduced frequency, hence obtaining sufficient material is not feasible.

*2) Examine whether nucleosome redistribution dynamics during the HS memory phase are affected by swi/snf or iswi mutations.*

This revision concerns the nucleosome redistribution dynamics after heat shock in the swi/snf and iswi mutants. As argued in detail above, we are unfortunately not able to perform these experiments at the moment due to inherent limitations of the biological material. (Our MNase-qPCR protocol requires at least one gram of starting material.) However, to determine whether loss of FGT1 and of SWI/SNF or ISWI remodelers have similar effects on nucleosome distributions at HS memory genes, we have assayed nucleosome occupancy in fgt1-1, brm-1 and chr11/chr11 chr17/+ plants at the rosette stage under normal growth conditions, where sufficient material could be obtained. The results are shown in Figure 8—figure supplement 2. Consistent with the expectation from the ChIP-seq analyses, which show binding of FGT1 and BRM to memory-gene promoters even in the absence of heat stress, our results demonstrate very similar changes in nucleosome occupancy in all three mutants relative to their respective wild-type. Namely, nucleosome occupancy was increased at *HSA32, HSP22* and *HSP18.2* in the mutants. Also, the *HSA32* profile in the Col wildtype with or without the *HSA32::LUC* transgene (parent and wild type, respectively) looked very similar, lending further support to the notion that the transgene behaves similarly to the endogenous *HSA32*. These results strengthen our interpretation that FGT1, BRM and ISWI interact functionally at the promoters of HS memory genes.

To better illustrate the role of FGT1 in nucleosome recovery after HS, we have calculated the relative nucleosome recovery rates based on the data shown in Figure 8 (see new Figure 8—figure supplement 1). Over the three analysed memory genes, the results clearly indicate that the +1 nucleosome recovers faster to pre-HS levels in *fgt1-1*. In contrast, nucleosome recovery dynamics were not affected at the non-memory gene *HSP101*.Taken together, these findings show that *fgt1* and the remodeler mutants show both a modified nucleosome profile before HS, and a specific defect during the memory phase in *fgt1*.

*3) Include as a control analysis of nucleosome occupancy of a gene that is not a target of FGT1 nor persistently expressed (i.e. HSP101).*

We have included nucleosome occupancy at the HS-inducible non-memory gene *HSP101* in all analyses of nucleosome occupancy (Figure 8 and Figure 8—figure supplement 1 and Figure 8—figure supplement 2). We have also included *HSP101* in the new rosette leaf analysis of *fgt1-1* and the remodeler mutants under non-stress conditions (Figure 8—figure supplement 2). In all cases, we do not see any deviations in the profile observed in the mutants compared to wild type.

In addition, we have included the HS-inducible non-memory genes *HSP70* and *HSP101* in the analysis of spliced and unspliced transcript levels presented in Figure 1—figure supplement 3. Here again, no difference was observed in *fgt1-1* compared to the parent.

Together, these new results corroborate the notion that the effects observed in *fgt1* and the remodeler mutants are specific for memory genes.

*4) Provide a better control for FGT1-H3 interaction.*

To independently support the interaction of FGT1 with histone H3, we have performed an in vivoanti-YFP immunoprecipitation experiment on the FGT1-YFP expressing line, followed by Western blotting with an anti-histone H3 antibody. The results of this experiment shown in the (new) Figure 2 demonstrate that FGT1-YFP can precipitate histone H3 from plant extracts, whereas overexpressed YFP alone cannot. While we cannot formally exclude that the interaction in the in vitropull-downs (now displayed as Figure 2—figure supplement 3) is mediated by the FLAG-tag and the in vivoprecipitation is due to FGT1 binding to other proteins, the most parsimonious and to us most plausible interpretation of these results is that FGT1 directly binds histone H3, consistent with the well-established activity of PHD domains as histone H3-methyl binding domains.

*Reviewer #1:*

In this manuscript, the authors identified FORGETTER1 (FGT1) as a new factor that is required for heat stress (HS) memory. […]

*1) The interaction between FGT1 and SWI/SNF and ISWI was shown by IP Mass-spec and BiFC experiments. As this is one of the key points of this paper, additional approaches should be employed to demonstrate the interaction.*

*2) The authors should test whether SWI/SNF and ISWI is required for FGT1 binding to HS memory genes or vice versa. This can be done by performing FGT1 ChIP analyses in the swi/snf or iswi mutants and SWI/SNF and ISWI ChIP in the fgt1 mutant.*

*3) The authors should examine whether nucleosome redistribution dynamics during the HS memory phase are affected by swi/snf or iswi mutations.*

*4) It would be also interesting to examine which histone marks are affected by fgt1 during HS memory.*

We agree with the reviewer that knowledge about potential histone marks affected by FGT1 would be very interesting. However, given the many possibilities, we feel it is outside of the scope of the present work to address this question.

*Reviewer #2:*

*To validate these observations, the authors measured the mRNA levels of both genes that exhibit persistent expression after heat stress (HSA32, HSP21, HSP22.0 and HSP18.2) and a gene that is induced by heat stress but does not show persistent expression (HSP101) in the fgt1-1 mutant. Surprisingly, despite having used an HSA32 reporter for the screen, the fgt1-1 mutant plants showed normal persistent expression of HSA32 and HSP18.2 and slightly defective persistence of HSP21 and HSP22.0. However, the authors claim that the very slight decrease in HSA32 after 45 and 69h are the source of the phenotype. Because there was no statistical analysis of this difference, it is unclear if it is significant.*

We have added statistics for the transcript level analysis shown in Figure 1, which demonstrates that the premature decline of both spliced and unspliced (where analysed) transcript levels of *HSA32, HSP18.2, HSP21* and HSP22.0 during the later timepoints is significant and is specific for memory genes (see also new Figure 1—figure supplement 3).

*These differences do not appear to be as strong as the differences observed with the Luciferase reporter (Figure 1). Quantitation of the luciferase for comparison would have clarified the phenotype.*

We have also added *LUC* quantitation in Figure 1—figure supplement 3 to indicate that the transgene behaves similar to the endogene. This is entirely consistent with the result for unspliced *HSA32*, where the primers do not discriminate between endogene and transgene.

*To convince the reader that the mutation is impacting transcription, ChIP against RNAPII would have helped.*

As described above, we were unfortunately not able to establish RNA PolII ChIP in our lab. Instead, we added data for unspliced transcripts of *HSP21, 101* and *70* to support our conclusions (see above).

*Furthermore, inclusion of the HSFA2 mutant for comparison would have made the strength of the phenotype easier to assess. It is possible that the mutation has a larger effect on the reporter and that the survival phenotype is the result of an effect on a different subset of genes (i.e., HSP21 and HSP22). If so, then the persistent expression of genes in response to heat stress involves both an FGT1-dependent and an FGT1-independent mechanism.*

To estimate the strength of the *fgt1-1* phenotype at the physiological level, we have included the *hsa32* mutant in Figure 1 and Figure 2-D. Previous publications report a similar strength in phenotype of *hsa32* and *hsfa2* (e. g. Charng et al., Plant Physiol 2007). In comparison, the phenotype of *fgt1-1* is weaker, suggesting additional factors contribute to the memory. We have clarified the conclusion that the phenotype of *fgt1-1* is caused by the cumulative misregulation of several memory genes and not just *HSA32*.

*The authors hypothesize that FGT1 promotes persistent expression of HSP21 and HSP22.0 by remodeling nucleosomes because the protein possesses a PHD finger domain and a Helicase C domain. Consistent with this hypothesis, FGT1 localizes to the nucleus, interacts with both active genes and heat stress-induced genes and physically interacts with the BRM (SWI/SNF) and CHR11/CHR17 (ISWI). Unfortunately, a number of experiments create uncertainty about this conclusion. First, the changes in nucleosome occupancy in the fgt1 mutant do not correlate with binding of FGT1: FGT1 binds strongly to the HSA32 promoter/TSS upon heat stress and this correlates with decreased nucleosome occupancy. The fgt1 mutant shows lower nucleosome occupancy prior to heat stress and this decreases to a comparable level to the wild type in the days following heat stress. Therefore, FGT1 is not required for the decrease in nucleosome occupancy at HSA32. In the cases of HSP18.2 and HSP22, although the fgt1 mutant shows increased nucleosome occupancy, the changes in the wild type strain upon heat stress are difficult to appreciate and, as mentioned above, there is no obvious effect of this mutation on the expression of HSP18.2. It would have been helpful to include a gene that is neither a target of FGT1 nor persistently expressed (i.e. HSP101) as a more relevant control. Likewise, as mentioned above, examining the effect of the HSFA2 mutant in this assay would have been informative.*

We show that both FGT1 and BRM are associated with memory genes already before HS (Figure 7). Consistently, there are changes in the nucleosome profiles already before HS. This is confirmed in the newly included analysis of *fgt1-1, brm-1* and *chr11/chr11 chr17*/+ rosette leaves presented in Figure 8—figure supplement 2. Upon HS, there is a decrease in nucleosome occupancy in both wild type and mutant. As the first timepoint analyzed was 4 h after the end of the HS, we are unable to say whether there are subtle differences in the immediate response. However, the recovery to pre-stress levels occurred faster in the *fgt1* mutant. To clarify this observation, we have calculated the rate of recovery at the peak nucleosome after HS relative to the respective NHS control (Figure 8—figure supplement 1). Again, this effect was not found in *HSP101*.

We agree with the reviewer that the analysis of nucleosome dynamics in *hsfa2* would be interesting. However, we feel it is outside of the scope of the present work to address this question.

*Also, peptide binding experiments show that the PHD finger interacts non-specifically with (non-overlapping) histone peptides. The authors did not include a biologically irrelevant peptide control, but the low affinity and the lack of specificity cast doubt on the claim that "FGT1 binds to the N-terminal region of H3[…]". It seems equally plausible that FGT1 binds to the FLAG peptide.*

Please refer to response to essential revision #4 above.

*Finally, ChIP seq shows that FGT1 associates with both the transcription start sites and the transcription termination sites of active genes. Given the DExD-like helicase domain, this raises the alternative hypothesis that the protein is involved in post-transcriptional events and might impact the ability of plants to survive through influencing protein levels.*

As seen from the representative screenshots in Figure 7—figure supplement 1 and the overall occupancy analysis (Figure 4, Figure 7), the peak of the association with FGT1 is upstream of the TSS and downstream of the TTS, and thus outside the transcribed region, which is not consistent with a role in RNA processing. Transcript levels of unspliced and spliced RNA were in good accordance, suggesting that processing was not affected. While we cannot formally exclude a role in regulating protein levels, we find that the most parsimonious interpretation of the results is that FGT1 functions in nucleosome positioning alongside remodeling complexes.

*The authors also explored the role of BRM and CHR11/CHR17 in heat stress memory. Mutations in BRM1 and FGT1 produced synthetic flowering and development phenotypes, suggesting non-overlapping roles in these processes. Both mutant plants were somewhat defective in heat stress protection after 3d of recovery. However, these mutant plants also showed significantly lower peak levels of the HS-induction of HSA32, HSP18.1, HSP21 and HSP22. Therefore, the decrease in expression afterward appears proportional to the wildtype, but starting at a significantly lower peak level. If so, then poor survival after 3d of recovery is likely due to lower levels of induced transcription, not less sustained transcription.*

We agree with the reviewer (as we mentioned in the original version) that the induction of memory genes is slightly altered in the remodeler mutants. Given the wide genome-wide distribution of the remodelers, clearly they have functions beyond HS memory. However, we did not find a defect in the acquisition of thermotolerance or basal thermotolerance (Figure 6—figure supplement 1–Figure 6—figure supplement 2). The interaction with FGT1, the defect in the physiological memory assay and the very similar nucleosome profiles in rosette leaves, together indicate a function in HS memory together with FGT1.

*Reviewer #3:*

*How plants adapt to heat stress and remember previous heat shocks is a very interesting area of biology. This paper breaks new ground by discovering a new component, FORGETTER1, which the authors show is necessary for plants to "remember" heat stress correctly at the level of transcriptional regulation. Interestingly, the fgt1-1 mutants have a specific defect in acquired thermotolerance, consistent with the proposed role for this gene in controlling the transcriptional response to temperature. The authors present a rather complete story, since they go from a genetic screen, to mapping and examining the underlying molecular mechanism. It is a bit of an open question as to the actual role of FGT1 in transcriptional memory itself, but that is probably beyond the scope of this study.*

*Here are a few suggestions that came to my mind on reading the paper, and resolving these might make the study even stronger:*

*1) For the ChIP-seq of FGT1 it would be nice to see screen shots of actual loci from a browser. Averaged data is rather hard to interpret. Relating to this, different gene lists were used for "active" genes and HS responsive genes for Figure 4. Isn't there a more relevant list of genes whose transcription appears to show "memory" of heatshock? Another way of looking at this, what about the transcriptome of fgt1 itself?*

In the new Figure 7, we show the association of FGT1 and BRM with early and late HS-inducible and memory genes. The HS gene classes are based on the microarrays that we published in Stief et al. 2014. They show a remarkable overlap of BRM and FGT1 for HS memory genes under control conditions and after ACC, respectively.

To improve the clarity of the presentation, we have included screen shots from the FGT1 and the BRM ChIP-seq experiments for several loci (*HSA32, HSP18.2, HSP21, HSP22, ACT7, HSP101*) that further illustrate the very similar binding patterns of the two proteins (Figure 7—figure supplement 1).

*2) The model for FGT1 action is interesting. It seems to invoke a specific role for +1 nucleosomes in responding to temperature. It's striking that there is such a strong enrichment for FGT1 at the +1 position as well as the TTS. Of course these are sites where there is global enrichment for H2A.Z-Nucleosomes (Coleman-Derr and Zilberman 2012) and H2A.Z-nucleosomes are evicted in response to temperature (Kumar and Wigge 2010). It is therefore curious that no attempt is made to synthesise what is already known about temperature-dependent gene expression and its regulation by chromatin. At a minimum, one could simply overlay the distribution of FGT1 with that of H2A.Z.*

We agree that a role for H2A.Z eviction in heat-shock memory related to FGT1 function represents an attractive possibility. Based on individual-locus genome browser analysis, there indeed seems to be some overlap between FGT1 and H2A.Z bound sequences. However, we have repeatedly assayed the *arp6* mutant, which is defective in H2A.Z incorporation into nucleosomes, in our HS memory assays, and did not observe any difference in HS memory. Thus, we conclude that FGT1 likely functions independently of H2A.Z.